# Breaking the Time-Frequency Granularity Discrepancy in Time-Series Anomaly Detection

## ABSTRACT

In light of the remarkable advancements made in time-series anomaly detection (TSAD), recent emphasis has been placed on exploiting the frequency domain as well as the time domain to address the difficulties in precisely detecting *pattern-wise* anomalies. However, in terms of anomaly scores, the *window granularity* of the frequency domain is inherently distinct from the *data-point granularity* of the time domain. Owing to this discrepancy, the anomaly information in the frequency domain has not been utilized to its full potential for TSAD. In this paper, we propose a TSAD framework, ***Dual-TF***, that simultaneously uses both the time and frequency domains while breaking the *time-frequency granularity discrepancy*. To this end, our framework employs *nested-sliding windows*, with the outer and inner windows responsible for the time and frequency domains, respectively, and aligns the anomaly scores of the two domains. As a result of the high resolution of the aligned scores, the boundaries of pattern-wise anomalies can be identified more precisely. In six benchmark datasets, our framework outperforms state-of-the-art methods by 12.0–147%, as demonstrated by experimental results.

## CCS CONCEPTS

• **Information systems** → *Temporal data*; **Data stream mining**; • **Computing methodologies** → **Spectral methods**; **Anomaly detection**; • **Mathematics of computing** → *Time series analysis*.

## KEYWORDS

Temporal domain, Frequency/Spectral domain, Granularity discrepancy, Outlier, Pattern anomaly

**ACM Reference Format:**

Anonymous Author(s). 2023. Breaking the Time-Frequency Granularity Discrepancy in Time-Series Anomaly Detection. In *Proceedings of The Web Conference (WWW '24)*. ACM, New York, NY, USA, 17 pages. https://doi.org/XXXXXXX.XXXXXXX

## 1 Introduction

Time series, which is ubiquitous in various Web-based contexts such as Web servers and cloud services, is a fundamental resource for analyzing Web traffic patterns. *Time-series anomaly detection (TSAD)* is usually formulated as identifying the data points that significantly diverge from the normal or usual behavior. TSAD is commonly used to monitor states in many Web-related domains (e.g., cloud

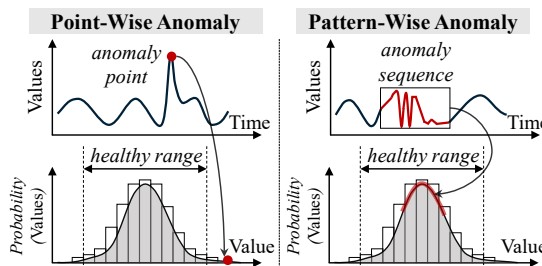

**Figure 1: Difficulty of detecting a pattern-wise anomaly (right) compared with a point-wise anomaly (left).**

services [45]) as well as manufacturing, healthcare, finance, energy, and environment [16, 33].

Time-series anomalies are mainly categorized into *point-wise* and *pattern-wise* (or *collective*) anomalies [17, 40], each of which is specified for a particular point and sequence. According to the behavior-driven taxonomy [32], the pattern-wise anomalies are further divided into *shapelet*, *seasonality*, and *trend* anomalies. Detecting pattern-wise anomalies is considered more difficult than detecting point-wise anomalies. As shown in Figure 1, a point-wise anomaly has a very unusual value deviating from the normal range of the probability distribution, whereas a pattern-wise anomaly may still have usual values that fall in the normal range [42].

In order to enhance the capability of capturing pattern-wise anomalies, recent studies have notably started considering both the time and frequency domains [56, 60, 64]. The former represents the values as a function of time, while the latter represents the periods (or cycles) as a function of frequency (see Appendix A for details). These recent approaches exploit the time domain mainly for finding point-wise anomalies and the frequency domain mainly for finding pattern-wise anomalies. The *uncertainty principle* for time-series representation [23], which can be taken to mean that if a particular anomaly is well represented in one domain, the anomaly may not be well represented in the other domain, provides strong support for this family of approaches.

Fully taking advantage of both domains for TSAD is, however, very challenging. Anomalies are specified at timestamps—i.e., in the time domain. Thus, the anomaly information found in the frequency domain needs to be aligned to the time domain for use in TSAD. Even worse, the *finest* granularity of the anomaly information in the frequency domain is coarser than that in the time domain. In detail, an anomaly score (*f-anomaly score*) in the frequency domain needs to be defined for a *window* (a sequence of data points), because a frequency spectrum can only be derived from a window (not a data point); in contrast, an anomaly score in the time domain (*t-anomaly score*) can be defined for a *data point*, e.g., reconstruction error [1, 49, 58, 63] and association discrepancy [55]. This problem is named the *time-frequency granularity discrepancy*.

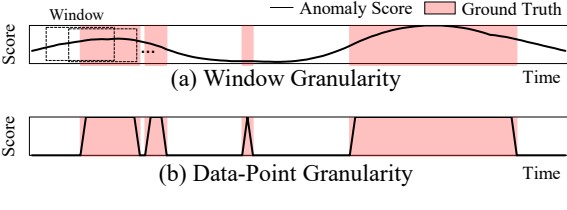

Figure 2: Comparison of ideal anomaly scores in (a) window granularity (existing) and (b) data-point granularity (ours).

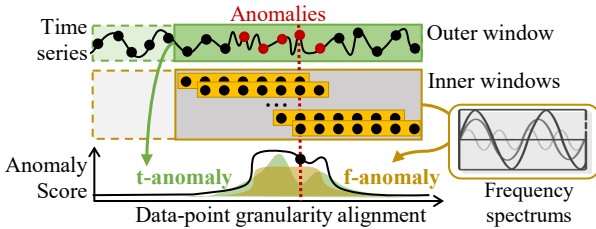

Figure 3: Our NS-windowing technique.

To resolve this discrepancy, existing approaches (e.g., TFAD [59]) sacrifice the data-point granularity even for the time domain and stick to the window granularity for both domains. That is, in both domains, an anomaly score is assigned to a window, and all data points in the window share the same score. Even with sliding windows, sharing the same score with all data points in a window significantly degrades the resolution of detecting pattern-wise anomalies. Especially when a window contains both normal and anomalous data points in Figure 2(a), the window granularity scheme inevitably fails to detect the *exact boundaries* of pattern-wise anomalies, thereby resulting in low overall accuracy.

In this paper, we propose a novel TSAD method, *Dual-TF*, which exploits both the time and frequency domains *without* the time-frequency granularity discrepancy. Our key solution is to use the *nested sliding window (NS-window)* to accommodate both time and frequency information while aligning them in the data-point granularity. As shown in Figure 3, for calculating t-anomaly scores, the outer window slides as usual to capture various time contexts. For calculating f-anomaly scores, the inner window slides only within the corresponding outer window to produce multiple frequency spectrums for each data point; these diverse frequency spectrums are compared with one another to return f-anomaly scores. Finally, these multiple t-anomaly scores and f-anomaly scores are consolidated for each data point to satisfy the data-point granularity. As a result, the boundaries of pattern-wise anomalies are more clearly and precisely identified in Figure 2(b).

Meanwhile, deep neural networks (DNNs) have demonstrated their capability to recognize intricate correlations within complex data characterized by large volume and dimensionality over the last decade. This trend has extended to multivariate time-series anomaly detection, resulting in an explosion of DNN-based methods suggesting methodological advances and improved performance [6, 38]. Notably, attention-based models, such as Transformer, offer the benefit of considering sequence dependencies and outperform previous state-of-the-art methods by a significant margin in time-series analysis [51]. The capability to effectively identify and analyze complex patterns and correlations in data is the reason for using the Anomaly Transformer [55] as the backbone.

*Dual-TF* includes two Anomaly Transformers for calculating the t-anomaly and f-anomaly scores, respectively. These scores are calculated and combined using the proposed NS-windowing scheme. Through the extensive comparison with ten TSAD methods for six datasets, *Dual-TF* is shown to improve the TSAD accuracy by 12.0–147%. Furthermore, consistent with our expectation, *Dual-TF*'s higher ability to capture the boundaries of pattern-wise anomalies

is visually confirmed, thus explaining the overall accuracy improvement. The idea of *using the NS-window for combining both domains* is very intuitive and widely applicable to any TSAD method based on the sliding window. We believe that the *simplicity* of our approach is a strong benefit because simple algorithms often make a big impact and gain widespread acceptance [48].

## 2 Related Work

### 2.1 Time-Series Anomaly Detection (TSAD)

The majority of TSAD methods are designed for unsupervised learning owing to a lack of anomaly labels. Traditional TSAD methods can be classified into statistical [10, 13, 46] and machine learning-based methods [19, 47]. In recent years, many studies have adopted deep learning, which is typically superior to traditional machine learning. Forecasting-based [18, 26, 61] and reconstruction-based methods are two well-known approaches. The former uses prediction errors as anomaly scores, while the latter uses reconstruction errors. Previous research has shown that reconstruction-based methods generally outperform forecasting-based methods [21, 62].

BeatGAN [63] is a reconstructive approach based on a generative adversarial network (GAN), which uses time-series warping for data augmentation to improve accuracy. MSCRED [58] exploits an attention-based ConvLSTM to account for temporal dependency. Autoencoder models are similarly employed for reconstruction in OmniAnomaly [49] and USAD [5]. RANSynCoders [1] improves autoencoder training efficiency via feature synchronization, bootstrapping, and quantile loss. Notably, a new reconstructive approach that combines series and prior association to make anomalies distinctive is proposed as *Anomaly Transformer* [55].

### 2.2 Frequency Domain Analysis for Time Series

Several recent methods use both the time and frequency (i.e., spectral) domains [57]. For forecasting, DEPTS [20] models complex periodicities with a learnable cosine function on top of residual learning. Autoformer [53] uses the auto-correlation generated via fast Fourier transform for long-term forecasting. FEDFormer [64], targeting long-term forecasting, extracts important frequency components using the frequency-enhanced block and frequency-enhanced attention. For unsupervised representation learning, CoST [52] has separate trend and seasonal encoders and compares the amplitude and phase of each sample to compute the time and frequency domain contrastive losses. BTSF [56] conducts iterative bilinear temporal-spectral fusion, where information on each domain is conveyed to a bilinear feature to model time-frequency dependencies.

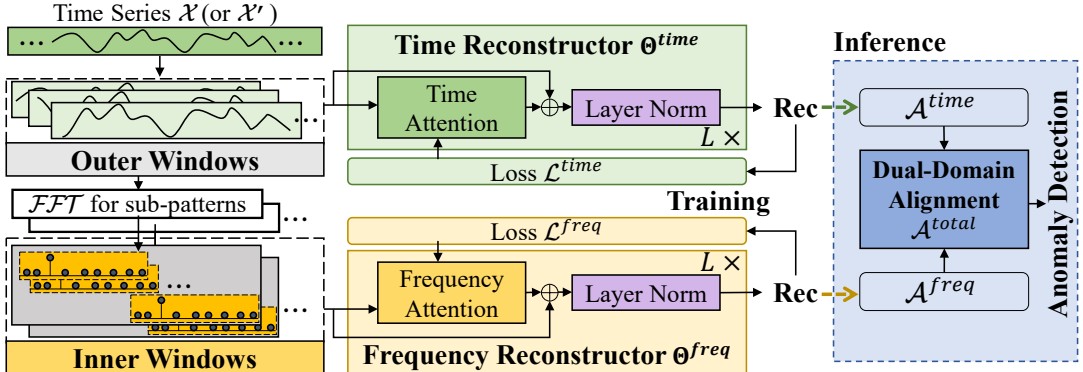

**Figure 4: Overview of the _Dual-TF_ framework for the training and inference phases.**

TF-C [60] augments the time and frequency domains independently to produce positive samples, followed by regularization to ensure coherence in the representation of both domains. However, these studies primarily focus on capturing _general_ time-series features and thus are unsuitable for detecting time-series anomalies. To identify pattern-wise features that can be defined only in the frequency domain, it is necessary to acquire frequency information that is appropriate for anomaly detection, as opposed to general frequency information obtained in the existing studies.

### 2.3 Frequency Domain Analysis for TSAD

Frequency-based models for TSAD have received much attention in recent years. The spectral residual (SR) introduced in SR-CNN [45] uses a frequency-based technique to generate a saliency map for TSAD. PFT [44] is a partial Fourier transform that achieves a speedup of an order of magnitude without sacrificing accuracy. TFAD [59] utilizes frequency domain analysis for TSAD with augmentation and decomposition. To the best of our knowledge, TFAD is the closest to our work since it uses the time and frequency domains together. Nevertheless, the existing methods (including TFAD) do _not_ offer the data-point granularity for the frequency domain, still facing challenges in precisely detecting pattern-wise anomalies.

### 3 TSAD Framework: _Dual-TF_

**Problem Formulation:** Let's consider a multivariate time series, $\mathcal{X} = \{x_1, x_2, \ldots, x_n\}$ ($\mathcal{X} \in \mathbb{R}^{n \times d}$), where $n$ is the number of data points and $d$ is the number of features. Using $\mathcal{X}$ as a training set, we aim at building an anomaly detector $\mathcal{A}(x_t, \Theta^{time}, \Theta^{frequency})$ that returns an anomaly score for a given data point using deep neural networks (e.g., autoencoders and Transformers) parameterized by $\Theta^{time}$ and $\Theta^{frequency}$ respectively for the time and frequency domains. Then, given another multivariate time series as a test set, $\mathcal{X}' \in \mathbb{R}^{n' \times d}$, the anomaly detector $\mathcal{A}()$ is used to classify each data point $x'_t$ in $\mathcal{X}'$ as being either normal or anomalous. Our problem is categorized as _unsupervised_ anomaly detection because no label information is used.

**Window Specification:** For the NS-windowing scheme in Figure 3, an outer window and its inner windows are continuously extracted from a multivariate time-series $\mathcal{X}$ (or $\mathcal{X}'$). An _outer window_ at timestamp $t$ is defined as $OW_t = \{x_t, x_{t+1}, \ldots, x_{t+w^{outer}-1}\}$

of length $w^{outer}$. Then, the original time series $\mathcal{X}$ (or $\mathcal{X}'$) is reorganized as a sequence of overlapping outer windows, $\mathcal{OW} = \{OW_1, OW_2, \ldots, OW_{n-w^{outer}+1}\}$[1]. For a given outer window $OW_t$, its _inner window_ of length $w^{inner}$ at timestamp $i \in [t, t + w^{outer} - w^{inner}]$ is defined $IW_i = \{x_i, x_{i+1}, \ldots, x_{i+w^{inner}-1}\}$. Then, again, an outer window is reorganized as a sequence of overlapping inner windows $\mathcal{IW}_t = \{IW_t, IW_{t+1}, \ldots, IW_{t+w^{outer}-w^{inner}}\}$[1].

**Time-Frequency Granularity Discrepancy:** The number of _degrees of freedom_ represents the number of values that can vary freely [12]. If the anomaly scores satisfy the data-point granularity, the degree of freedom should be the length of a time series $n$; on the other hand, if the anomaly scores follow the window granularity, the degree of freedom should be much smaller than $n$ because the scores cannot vary within the same window. Therefore, the _time-frequency granularity discrepancy_ is formally defined as when d.f.({t-anomaly scores}) ≠ d.f.({f-anomaly scores}), where d.f.(·) denotes the degree of freedom.

### 3.1 Overall Framework of _Dual-TF_

Figure 4 shows the overall procedure of _Dual-TF_. By the scheme of NS-windowing, a multivariate time series $\mathcal{X}$ (or $\mathcal{X}'$) is transformed into a sequence of outer windows and a sequence of inner windows for each outer window. _Dual-TF_ employs _two_ neural network reconstructors: the _time reconstructor_ and the _frequency reconstructor_, each for time and frequency domain. The outer windows are fed to the time reconstructor. On the other hand, a fast Fourier transform (FFT) [39] converts each inner window from its original time domain to a representation in the frequency domain; then, the converted inner windows are fed to the frequency reconstructor.

These reconstructors are _individually_ trained to minimize the reconstruction losses, following the conventional procedure of reconstruction-based TSAD [1, 50, 55]. The batches for the two reconstructors should be constructed separately due to the different dimensionalities of the inputs to the two reconstructors. During the inference phase, an anomaly score for each data point is calculated using the reconstruction losses from the two reconstructors. To break the time-frequency granularity discrepancy and thus achieve the data-point granularity in both domains, we align the

---

[1]The slide step is set to be 1 for finding the boundaries of pattern-wise anomalies more precisely as well as for simplifying the expression.

time-domain reconstruction losses from an outer window with the frequency-domain reconstruction losses from its inner windows.

## 3.2 Time-Frequency Dual-Domain Training

**Optimal Window Length:** For the NS-windowing scheme, one of the most important issues is to determine the proper window lengths, $w^{outer}$ and $w^{inner}$. Because the primary goal of using inner windows is to capture pattern-wise anomalies, we set them to embody major periods based on spectral analysis. In detail, each dimension of the entire time series $X$ is converted to the frequency domain by the fast Fourier transform (FFT). The output of the FFT is Hermitian-symmetric; that is, the positive-frequency terms are the complex conjugates of the corresponding negative-frequency terms. The negative frequency terms can be ignored because of the redundancy. Here, a frequency *magnitude* is defined as the absolute value (or modulus) of a complex number. Then, the most *dominant* frequency whose magnitude is the largest in each dimension is identified; the smallest dominant frequency in all dimensions is chosen as the *representative* frequency $v^{major}$ in $X$. Last, we set $w^{inner} = \lceil \frac{1}{v_{major}} \rceil$, because the period is the reciprocal of the frequency, and $w^{outer} = \rho \cdot w^{inner}$, where $\rho$ ($> 1$) is a hyperparameter. See Appendix B for the details of the window-length selection.

We briefly discuss the optimality of the inner windows. The time-frequency uncertainty principle [22] states that the exact time and frequency of a signal can never be known simultaneously [25]. In determining the window size, this principle indicates the trade-off between time and frequency uncertainty within a window. In the time domain, the smaller $w^{inner}$, the lower the uncertainty; in the frequency domain, the smaller $w^{inner}$, the greater the uncertainty. Let $\mathcal{U}^{time}(w^{inner})$ and $\mathcal{U}^{freq}(w^{inner})$, respectively, denote the uncertainty in the time and frequency domains, given a window of length $w^{inner}$. (See Appendix C the formal definition of the uncertainty.) Then, the uncertainty in the two domains is

$$\mathcal{U}(w^{inner}) = \mathcal{U}^{time}(w^{inner}) + \mathcal{U}^{freq}(w^{inner}). \quad (1)$$

Theorem 3.1 formally states the optimal condition for the inner window length $w^{inner}$.

THEOREM 3.1 (OPTIMAL WINDOW LENGTH). *When* $w^{inner} = \lceil \frac{1}{v_{major}} \rceil$, *the uncertainty within the window,* $\mathcal{U}(w^{inner})$ *in Eq.* (1), *is minimized.*

PROOF. See Appendix C for the proof. □

**Reconstructor Network:** While any neural network reconstructor is applicable to *Dual-TF*, we choose Anomaly Transformer [55] because it has shown the state-of-the-art performance. To make this paper be self-contained, we briefly describe the key mechanism of the Anomaly Transformer. Each input is $X^0 = OW_t$ for the time reconstructor and $X^0 = \mathcal{IW}_t$ for the frequency reconstructor. Following the self-attention mechanism, the output of the $l$-th layer ($l \in [1, n^{layer}]$) is defined by

$$Q, K, V = X^{l-1}W_Q^l, X^{l-1}W_K^l, X^{l-1}W_V^l$$

$$\text{Attention}(X^l) = \text{Softmax}(\frac{QK^\top}{\sqrt{d^{model}}}) V \quad (2)$$

$$\Theta^*(X^l) = \widehat{X^l} = \text{LayerNorm}(X^{l-1} + \text{Attention}(X^{l-1})),$$

where $Q$, $K$, and $V$ are queries, keys, and values; $W_Q^l$, $W_K^l$, and $W_V^l$ are the learnable parameters for them; $d^{model}$ is the number of hidden channels; and $\widehat{X^l}$ is the output of the reconstruction network. More importantly, the association discrepancy is defined as the KL-divergence between the prior association ($P^l$) and the series association ($S^l$),

$$AssDis(P, S; X) = \frac{1}{n^{layer}} \sum_{l=1}^{n^{layer}} (\text{KL}(P^l \| S^l) + \text{KL}(S^l \| P^l)). \quad (3)$$

where $P^l$ is generated by the Gaussian kernel to represent the adjacent context and $S^l$ is defined as the usual self-attention in Eq. (2) to represent the overall context.

The $n^{layer}$ layers of the Anomaly Transformer backbone in Eq. (2) are used for both reconstructors. For the time reconstructor, $Q, K, V \in \mathbb{R}^{w^{outer} \times d^{model}}$; $W_Q^l, W_K^l, W_V^l \in \mathbb{R}^{d^{model} \times d^{model}}$; $P^l, S^l \in \mathbb{R}^{w^{outer} \times w^{outer}}$. In the same manner, for the frequency reconstructor, $Q, K, V \in \mathbb{R}^{(m \times w^{inner}) \times d^{model}}$; $W_Q^l, W_K^l, W_V^l \in \mathbb{R}^{d^{model} \times d^{model}}$; $P^l, S^l \in \mathbb{R}^{(m \times w^{inner}) \times (m \times w^{inner})}$, where $m$ denotes the number of inner windows per outer window, i.e., $m = w^{outer} - w^{inner} + 1$.

**Time Reconstructor Loss:** Time reconstructor forms each outer window $OW_t = \{x_t, x_{t+1}, \ldots, x_{t+w^{outer}-1}\}$ ($OW_t \in \mathbb{R}^{w^{outer} \times d}$), where $t \in [1, n - w^{outer} + 1]$, to $\widehat{OW}_t = \{\hat{x}_t, \hat{x}_{t+1}, \ldots, \hat{x}_{t+w^{outer}-1}\}$ ($\widehat{OW}_t \in \mathbb{R}^{w^{outer} \times d}$) using the parameter $\Theta^{time}$. Then, the reconstruction loss of $OW_t$ is formulated by

$$RecLoss^{time}(OW_t, \widehat{OW}_t) = \sum_{i=t}^{t+w^{outer}-1} \|x_i - \hat{x}_i\|_2^2. \quad (4)$$

It is evident that the time domain satisfies the data-point granularity in Eq. (4). The association discrepancy is (optionally) added to the final loss,

$$\mathcal{L}^{time}(OW_t, \widehat{OW}_t) = RecLoss^{time}(OW_t, \widehat{OW}_t)$$
$$- \lambda \cdot AssDis^{time}(P, S; OW_t), \quad (5)$$

where $\lambda$ ($> 0$) is the hyperparameter for weighting the association discrepancy. These losses for the outer windows in a batch are summed up to update the parameter $\Theta^{time}$ via backpropagation.

**Frequency Reconstructor Loss:** A sequence of overlapping inner windows $\mathcal{IW}_t = \{IW_t, IW_{t+1}, \ldots, IW_{t+w^{outer}-w^{inner}}\}$ is derived, given an outer window $OW_t$. First, each inner window $IW_t$ ($\in \mathbb{R}^{w^{inner} \times d}$) is converted to a frequency spectrum $\mathcal{FFT}(IW_t)$ ($\in \mathbb{R}^{w^{inner} \times d}$), where $\mathcal{FFT}()$ returns a sequence of the magnitudes in the result of the FFT. That is, $\mathcal{FFT}(IW_t)$ is regarded as the *counterpart* of the data point $x_t$ in the frequency domain. Here, $\mathcal{FFT}()$ is separately applied to each of $d$ dimensions. Then, $\{\widehat{\mathcal{FFT}(IW_t)}, \widehat{\mathcal{FFT}(IW_{t+1})}, \ldots\}$ is reconstructed by the frequency reconstructor from $\{\mathcal{FFT}(IW_t), \mathcal{FFT}(IW_{t+1}), \ldots\}$. Last, the reconstruction loss of $\mathcal{IW}_t$ generated from $OW_t$ is defined as

$$RecLoss^{freq}(\mathcal{IW}_t, \widehat{\mathcal{IW}}_t)$$
$$= \sum_{i=t}^{t+w^{outer}-w^{inner}} \sum_{j=1}^{d} \|\mathcal{FFT}(IW_i)_{[j]} - \widehat{\mathcal{FFT}(IW_i)}_{[j]}\|_2^2. \quad (6)$$

Here, $\mathcal{FFT}()_{[j]}$ is the frequency spectrum of the $j$-th dimension. By virtue of the sequence of inner windows, the frequency

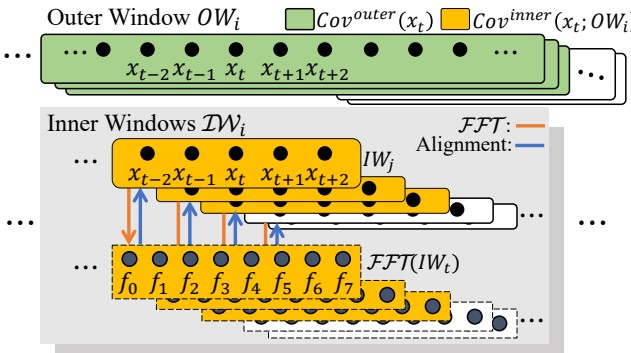

**Figure 5: Data-point granularity alignment.**

domain also satisfies the data-point granularity in Eq. (6) because an inner window is created for every data point except the last $w^{inner}$ ones within the outer window; we note that these uncovered data points will be covered soon by the succeeding outer windows. Similar to Eq. (5), for the update of the parameter $\Theta^{freq}$, the total loss is defined as

$$\mathcal{L}^{freq}(\mathcal{IW}_t, \widehat{\mathcal{IW}_t}) = RecLoss^{freq}(\mathcal{IW}_t, \widehat{\mathcal{IW}_t})$$
$$- \lambda \cdot AssDis^{freq}(P, S; \mathcal{IW}_t). \quad (7)$$

## 3.3 Data-Point Granularity Alignment for Anomaly Scoring

Using the two trained reconstructors $\Theta^{time}$ and $\Theta^{freq}$, we derive the anomaly score for each data point $x'$ in a test set $\mathcal{X}'$. First, an anomaly score in the time domain (*t-anomaly score*) and an anomaly score (*f-anomaly score*) in the frequency domain are derived separately from the two reconstructors. Then, the t-anomaly score and the f-anomaly score are combined to form the final anomaly score, which will be compared against a threshold.

Because an entire time series (or an outer window) is converted to a sequence of *overlapping* outer (or inner) windows, each data point is covered by *multiple* outer (or inner) windows. Definitions 3.2 and 3.3 formalize the sets of *covering* outer and inner windows for a given data point.

**Definition 3.2** (Covering Outer Window). Given a data point $x_t$, the set of *covering outer windows* is defined by $Cov^{outer}(x_t) = \{OW_i \mid x_t \in OW_i, 1 \le i \le n - w^{outer} + 1\}$.

**Definition 3.3** (Covering Inner Window). Given a data point $x_t$ and its covering outer window $OW_i \in Cov^{outer}(x_t)$, the set of *covering inner windows* is defined by $Cov^{inner}(x_t; OW_i) = \{IW_j \mid x_t \in IW_j \land IW_j \in OW_i, i \le j \le i + w^{inner} - 1\}$.

Figure 5 illustrates how t-anomaly scores and f-anomaly scores are calculated and then aligned. First, a ***t-anomaly score*** is easily calculated *for each data point within a given outer window* using the reconstruction loss. Though any reconstruction-based criterion is usable, we follow the one proposed for the state-of-the-art model, Anomaly Transformer [55],

$$\mathcal{A}^{time}(x_t; OW_i) = [\text{Softmax}(-AssDis^{time}(P, S; OW_i))$$
$$\odot RecLoss^{time}(OW_i, \widehat{OW_i})]_{x_t}, \quad (8)$$

where $AssDis()$ and $RecLoss()$ are the association discrepancy and the reconstruction loss used in Eq. (5), and $[]_{x_t}$ returns the value for $x_t$. Second, an ***f-anomaly score*** is calculated in two steps. **(i)** A score $\mathcal{A}^{freq}(IW_j; \mathcal{IW}_i)$ is derived *for each covering inner window within the context of all inner windows from the outer window*; that is, it basically represents the spectral difference of $IW_j$ to the other inner windows in $\mathcal{IW}_i$. **(ii)** The exponential function is applied to each $\mathcal{A}^{freq}(IW_j; \mathcal{IW}_i)$, and these results are averaged for the outer window, where $Cov^{inner} = Cov^{inner}(x_t; OW_i)$,

$$\mathcal{A}^{freq}(x_t; OW_i) = \frac{1}{|Cov^{inner}|} \sum_{IW_j \in Cov^{inner}} \exp\left(\mathcal{A}^{freq}(IW_j; \mathcal{IW}_i)\right),$$

$$\mathcal{A}^{freq}(IW_j; \mathcal{IW}_i) = [\text{Softmax}(-AssDis^{freq}(P, S; \mathcal{IW}_i))$$
$$\odot RecLoss^{freq}(\mathcal{IW}_i, \widehat{\mathcal{IW}_i})]_{IW_j}. \quad (9)$$

We note that an f-anomaly score is derived for each data point using its covering inner windows, where different windows cover different intervals in an outer window. By supporting the data-point granularity for f-anomaly scores, we want to distinguish between when $x_t$ is at the center of or near the boundary of a pattern-wise anomaly. As $x_t$ is located at a more central location of this anomaly, its set of covering inner windows overlap the anomaly more significantly, where each $\mathcal{A}^{freq}(IW_j; \mathcal{IW}_i)$ in Eq. (9) becomes higher. Thus, to make the f-anomaly score for the central data point *stand out*, we take the exponential of each $\mathcal{A}^{freq}(IW_j; \mathcal{IW}_i)$. This simple treatment for TSAD is proven to be very effective, as shown in the ablation study.

Next, Eqs. (8) and (9) are averaged for all of $x_t$'s covering outer windows, where $Cov^{outer} = Cov^{outer}(x_t)$

$$\mathcal{A}^{time \text{ or } freq}(x_t) = \frac{1}{|Cov^{outer}|} \sum_{OW_i \in Cov^{outer}} \mathcal{A}^{time \text{ or } freq}(x_t; OW_i), \quad (10)$$

and this score for each $x_t$ is min-max normalized with respect to the scores of all data points in $\mathcal{X}$.

At last, considering *both* the t-anomaly and f-anomaly scores, the final anomaly score is defined by

$$\mathcal{A}^{total}(x_t) = \mathcal{A}^{time}(x_t) + \mathcal{A}^{freq}(x_t). \quad (11)$$

**Remark 3.4.** *The t-anomaly and f-anomaly scores in Eq. (10)* **break** *the time-frequency granularity discrepancy.*

**Proof.** Given any pair of $x_i$ and $x_j$ where $i \ne j$, $Cov^{outer}(x_i) \ne Cov^{outer}(x_j)$ and $Cov^{inner}(x_i; \cdot) \ne Cov^{inner}(x_j; \cdot)$ by the definition of the NS-windows. Therefore, d.f.$(\{\mathcal{A}^{time}(x_t) \mid 1 \le t \le n)\})$ = d.f.$(\{\mathcal{A}^{freq}(x_t) \mid 1 \le t \le n)\})$ = $n$. □

## 4 Evaluation

For reproducibility, the source code of our framework is available at https://anonymous.4open.science/r/DualTF.

### 4.1 Experiment Settings

**Datasets:** Table 1 summarizes the benchmark datasets used in the experiments. Here, each anomaly is categorized as a pattern-wise anomaly if the length of its anomaly interval is greater than 1 and a point-wise anomaly otherwise [7]. The number in the parenthesis along the number of point-wise or pattern-wise anomalies is the ratio of the number of anomaly-labeled data points of a specific category to the total number of anomaly-labeled data points.

Table 1: Benchmark dataset statistics.

| Datasets | Applications | $w^{inner}$ | # Train | # Test | Entity×Dim. | # Point Anomaly (Ratio) | # Pattern Anomaly (Ratio) | Avg. Length of Pattern Anomaly |
|---|---|---|---|---|---|---|---|---|
| TODS (Point) | Synthetic | 25 | 20,000 | 5,000 | 2 × 1 | 250 (100%) | 0 (0%) | N/A |
| TODS (Pattern) | Synthetic | 25 | 20,000 | 5,000 | 3 × 1 | 0 (0%) | 250 (100%) | 10 |
| ASD | Server Monitoring | 288 | 8,527 | 4,320 | 12 × 19 | 0 (0%) | 199 (100%) | 31 |
| ECG | Medical Checkup | 143 | 6,995 | 2,851 | 9 × 2 | 0 (0%) | 208 (100%) | 208 |
| PSM | Server Monitoring | 360 | 132,481 | 87,841 | 1 × 25 | 16 (0.07%) | 24,365 (99.93%) | 435 |
| CompanyA | Server Monitoring | 144 | 21,600 | 13,302 | 3 × 8 | 10 (8.53%) | 104 (91.47%) | 4 |

Table 2: Performance comparison between TSAD methods in terms of the best point-wise $F_1$ score with the highest scores highlighted in bold.

| Methods | TODS (Point) | | TODS (Pattern) | | | ASD | ECG | PSM | CompanyA | Avg. ↑ | Rank ↓ |
|---|---|---|---|---|---|---|---|---|---|---|---|
| | Gloabl | Contextual | Shapelet | Seasonal | Trend | | | | | | |
| ISF | 0.943 (±0.017) | 0.164 (±0.000) | 0.103 (±0.000) | 0.093 (±0.000) | 0.209 (±0.000) | 0.295 (±0.000) | 0.256 (±0.000) | 0.478 (±0.000) | 0.134 (±0.000) | 0.297 (±0.002) | 9 |
| LOF | 0.933 (±0.000) | 0.093 (±0.000) | 0.096 (±0.000) | 0.092 (±0.000) | 0.093 (±0.000) | 0.376 (±0.067) | 0.327 (±0.000) | 0.524 (±0.099) | 0.059 (±0.019) | 0.288 (±0.010) | 11 |
| OCSVM | 0.937 (±0.019) | 0.170 (±0.000) | 0.104 (±0.000) | 0.093 (±0.000) | 0.094 (±0.000) | 0.266 (±0.071) | 0.264 (±0.000) | 0.469 (±0.000) | 0.266 (±0.019) | 0.296 (±0.011) | 10 |
| VAE | 0.915 (±0.031) | 0.584 (±0.034) | 0.503 (±0.094) | 0.847 (±0.017) | 0.181 (±0.002) | 0.327 (±0.022) | 0.274 (±0.003) | 0.443 (±0.000) | 0.261 (±0.028) | 0.482 (±0.014) | 4 |
| MS-RNN | 0.839 (±0.000) | 0.553 (±0.000) | 0.248 (±0.144) | 0.799 (±0.022) | 0.180 (±0.000) | 0.379 (±0.003) | 0.276 (±0.002) | 0.443 (±0.000) | 0.228 (±0.003) | 0.438 (±0.014) | 5 |
| OmniAnomaly | 0.543 (±0.001) | 0.542 (±0.008) | 0.149 (±0.004) | 0.203 (±0.017) | 0.185 (±0.013) | 0.197 (±0.096) | 0.216 (±0.037) | 0.467 (±0.098) | 0.182 (±0.046) | 0.298 (±0.009) | 8 |
| RANSynCoders | 0.674 (±0.127) | 0.482 (±0.000) | 0.166 (±0.002) | 0.163 (±0.007) | 0.175 (±0.011) | 0.383 (±0.234) | 0.208 (±0.003) | 0.571 (±0.017) | 0.112 (±0.027) | 0.326 (±0.026) | 7 |
| TranAD | 0.569 (±0.000) | 0.553 (±0.000) | 0.165 (±0.000) | 0.179 (±0.030) | 0.169 (±0.000) | 0.294 (±0.007) | 0.461 (±0.028) | 0.443 (±0.000) | 0.225 (±0.008) | 0.340 (±0.000) | 6 |
| TFAD | 0.878 (±0.000) | 0.871 (±0.009) | 0.558 (±0.150) | 0.854 (±0.018) | 0.363 (±0.001) | 0.432 (±0.003) | 0.356 (±0.002) | 0.537 (±0.080) | 0.276 (±0.071) | 0.569 (±0.035) | 3 |
| Anomaly Transformer | 0.943 (±0.000) | 0.942 (±0.000) | 0.730 (±0.000) | 0.867 (±0.028) | 0.460 (±0.005) | 0.425 (±0.017) | 0.464 (±0.001) | 0.578 (±0.001) | 0.317 (±0.099) | 0.636 (±0.011) | 2 |
| **Dual-TF** | **0.968** (±0.017) | **0.943** (±0.001) | **0.741** (±0.005) | **0.925** (±0.041) | **0.476** (±0.017) | **0.661** (±0.019) | **0.538** (±0.076) | **0.723** (±0.047) | **0.436** (±0.021) | **0.712** (±0.011) | 1 |

See Appendix D for the generation of the TODS dataset [32]. The ASD [34], ECG [28], and PSM [2] are public datasets commonly used for evaluating TSAD. The only proprietary dataset is the CompanyA (anonymized) dataset, which is derived from the operation of cloud servers and represents service traffic.

**Baselines:** We conduct a comparative analysis of *Dual-TF* against both traditional and recent works. For traditional methods, ISF [35], LOF [11], OCSVM [47], and Variational Autoencoder (VAE) [24] are included. For the state-of-the-art methods, Modified-RNN (MS-RNN) [30], OmniAnomaly [49], RANSynCoders [1], TranAD [50], TFAD [59], and Anomaly Transformer [55] are included. See Appendix D for the descriptions of the baselines.

**Evaluation Metrics:** To evaluate the detection accuracy at the data-point level, we primarily employ the *point-wise* $F_1$ score [3]. In this case, a predicted anomaly is valid if it falls within a small margin (i.e., ten timestamps) of the actual location of the anomaly. In contrast, it is well-known that the widely-used point-adjusted (PA) metric has overestimation issues [49, 54, 55], as a window is considered correct if both a predicted anomaly and the true anomaly occur just within the same window. Despite the fact that the PA metric is inappropriate for our work, we report the results using the PA metric in Appendix E so that they can be compared to the findings of other studies. In addition, we report the *best* $F_1$ score using the anomaly threshold that produces the highest $F_1$ score across all methods in order to eliminate the effect of threshold selection. We repeat every test *three* times with random seeds and report the average as well as the standard deviation.

Furthermore, addressing recent concerns on evaluation metrics [31, 41], we adopt new evaluation metrics designed for TSAD. The Range Area Under the Curve (R_AUC) and the Volume Under the Surface (VUS) [41] are employed, where the receiver operating characteristic (ROC) curve and precision-recall (PR) curves are considered. The VUS extends the mathematical model of the R_AUC by allowing the buffer length to be varied. Thus, R_AUC_ROC and R_AUC_PR are defined for the former, and VUC_ROC and VUC_PR are defined for the latter. These metrics can accurately evaluate the detection of pattern-wise (range) anomalies. Moreover, the VUS reduces the influence of an anomaly threshold.

**Table 3: Performance comparison for *Dual-TF* in terms of VUS [41] and other new evaluation metrics with the highest scores highlighted in bold. See [41] for the details of these evaluation metrics.**

| Evaluation Metrics | Methods | TODS (Point) | | TODS (Pattern) | | | ASD | ECG | PSM | CompanyA |
|---|---|---|---|---|---|---|---|---|---|---|
| | | Gloabl | Contextual | Shapelet | Seasonal | Trend | | | | |
| R_AUC_ROC | Anomaly Transformer | 0.9995 | 0.9859 | 0.8457 | 0.9272 | 0.6277 | 0.8498 | 0.6432 | 0.6158 | 0.8493 |
| | *Dual-TF* | **0.9998** | **0.9995** | **0.9097** | **0.9611** | **0.7035** | **0.9013** | **0.7216** | **0.7735** | **0.8653** |
| R_AUC_PR | Anomaly Transformer | 0.9994 | 0.9862 | 0.6878 | 0.7736 | 0.3713 | 0.5263 | 0.2447 | 0.4789 | 0.4139 |
| | *Dual-TF* | **0.9998** | **0.9996** | **0.7925** | **0.8719** | **0.4287** | **0.6058** | **0.3809** | **0.6304** | **0.4254** |
| VUS_ROC | Anomaly Transformer | 0.9354 | 0.9160 | 0.8065 | 0.9147 | 0.6222 | 0.7952 | 0.6343 | 0.6073 | 0.8335 |
| | *Dual-TF* | **0.9373** | **0.9322** | **0.8843** | **0.9380** | **0.6992** | **0.8505** | **0.7067** | **0.7752** | **0.8568** |
| VUS_PR | Anomaly Transformer | 0.8985 | 0.8765 | 0.6000 | 0.6920 | 0.3560 | 0.4466 | 0.2424 | 0.4665 | 0.3590 |
| | *Dual-TF* | **0.9053** | **0.9014** | **0.6950** | **0.7620** | **0.4016** | **0.5127** | **0.3754** | **0.6131** | **0.3694** |

**Table 4: Ablation study for *Dual-TF* in terms of the best point-wise $F_1$ score with the highest scores highlighted in bold.**

| Variations | TODS (Point) | | TODS (Pattern) | | | ASD | ECG | PSM | CompanyA | Average |
|---|---|---|---|---|---|---|---|---|---|---|
| | Gloabl | Contextual | Shapelet | Seasonal | Trend | | | | | |
| (i) w/o Time Reconstructor | 0.439 | 0.661 | 0.715 | 0.858 | 0.476 | 0.629 | 0.267 | 0.500 | 0.247 | 0.532 |
| (ii) w/o Frequency Reconstructor | 0.943 | 0.942 | 0.728 | 0.824 | 0.457 | 0.415 | 0.460 | 0.578 | 0.264 | 0.623 |
| (iii) w/o NS-Windowing | 0.388 | 0.600 | 0.691 | 0.798 | 0.213 | 0.313 | 0.263 | 0.397 | 0.229 | 0.433 |
| (iv) w/o Exponential Average | 0.953 | 0.905 | 0.620 | 0.780 | 0.465 | 0.577 | 0.507 | 0.677 | 0.322 | 0.645 |
| *Dual-TF* | **0.968** | **0.943** | **0.741** | **0.925** | **0.476** | **0.661** | **0.538** | **0.723** | **0.436** | **0.712** |

**Implementations and Model Configurations:** *Dual-TF* is implemented using PyTorch 1.13.1. The only hyperparameter $\rho$ for *Dual-TF*, which specifies the size of an outer window, is set to 2 for all datasets. The Adam optimizer is used with an initial learning rate of $10^{-4}$. The batch size is 4 considering the large size of each training instance and the memory budget of a GPU. The training process is early stopped within 10 epochs. Following the author implementation of Anomaly Transformer [55], the weight $\lambda$ in Eqs. (4) and (6) is 3, the number of layers $n^{layer}$ is 3, the number of hidden channels $d^{model}$ is 512, and the number of attention heads is 8. For all baseline methods, the authors' source code is employed without any modification, and the hyperparameters are set to be the default values in the author implementation. All experiments are conducted on a server equipped with an NVIDIA RTX 3090Ti. See Appendix D for details.

## 4.2 Overall Performance Comparison

Table 2 shows the best point-wise $F_1$ score of *Dual-TF* as well as all baselines for six benchmark datasets. *Dual-TF* is shown to significantly outperform the state-of-the-art TSAD methods; specifically, it yields a detection accuracy of 12.0–147% higher on average than the other methods. Anomaly Transformer and TFAD are ranked second and third, respectively. *Dual-TF* effectively handles both point-wise (in TODS (Point)) and pattern-wise (in TODS (Pattern), ASD, ECG, PSM, and CompanyA) anomalies while showing greater improvement in detecting the pattern-wise anomalies. Compared with Anomaly Transformer, the improvement in the $F_1$ score is 0.10–2.63% for point-wise anomalies, and it is *increased* to 1.55–55.4% for pattern-wise anomalies owing to the incorporation of the frequency domain. Additionally shown in Table 3, the improvement

in the range-AUC measures (R_AUC_ROC or R_AUC_PR) is up to 1.38% for point-wise anomalies, and up to 55.7% for pattern-wise anomalies. Moreover, the enhancement in the VUS-based measures (VUS_ROC or VUS_PR) is up to 2.84% for point-wise anomalies, and up to 54.8% for pattern-wise anomalies with the help of the frequency domain. Moreover, compared with TFAD, which uses both the time and frequency domains, the $F_1$ score improves by 8.25–57.9% because of the higher resolution achieved by the data-point granularity. Overall, these results indeed demonstrate the value of combining both domains while breaking the time-frequency granularity discrepancy.

## 4.3 Ablation Study

The contribution of each main component of *Dual-TF* to the anomaly detection accuracy is investigated through an ablation study in Table 4. Specifically, when (i) time reconstructor is removed, (ii) frequency reconstructor is removed in Figure 4, (iii) the window granularity is enforced for the frequency domain without the NS-windowing scheme, or (iv) the exponential average in Eq. (9) for aggregating f-anomaly scores is replaced with the arithmetic average, the $F_1$ score for each variation is measured. All of these main components are shown to be important, with the *NS-windowing* component having the most outstanding effect. Interestingly, the use of f-anomaly scores at the window granularity in the third variation may actually contaminate t-anomaly scores by unnecessarily increasing the final anomaly scores for a normality interval and thus producing a large number of false positives. Overall, this comprehensive ablation study reaffirms the significance of breaking the time-frequency granularity discrepancy.

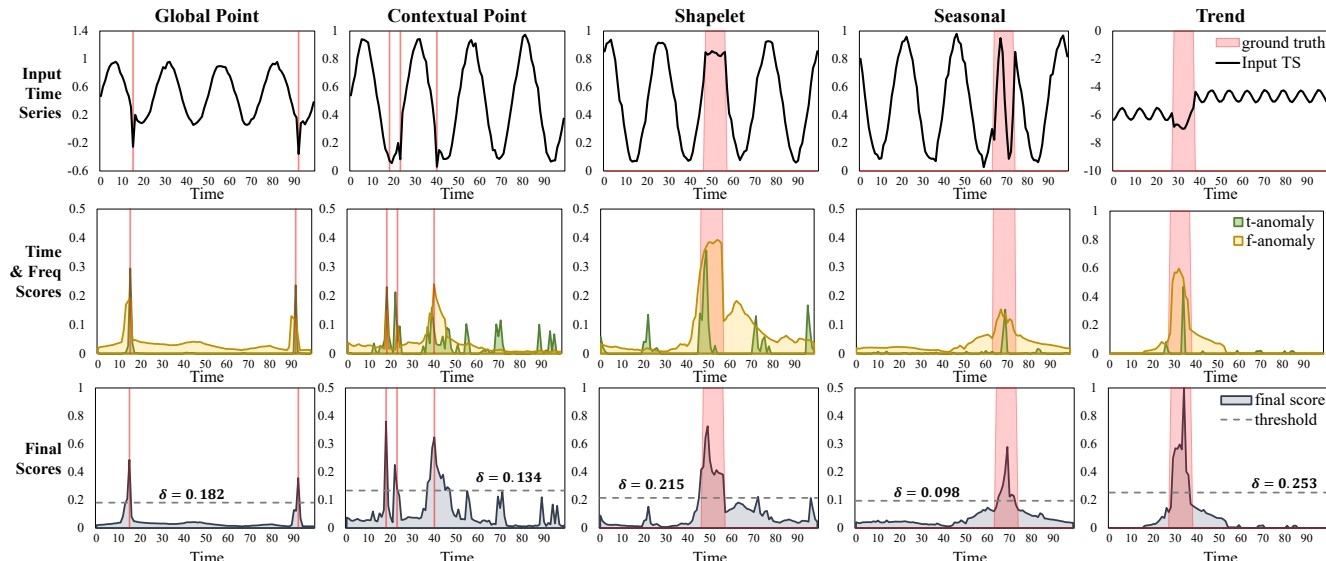

**Figure 6: Visualization of dual-domain anomaly scores from *Dual-TF* for different categories of point- and pattern-wise anomalies using the TODS dataset.**

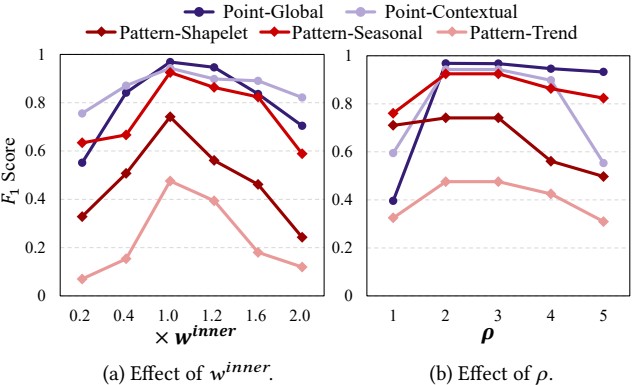

(a) Effect of $w^{inner}$.

(b) Effect of $\rho$.

**Figure 7: Effect of the inner and outer window lengths in the TODS dataset.**

## 4.4 Qualitative Analysis through Visualization

Figure 6 visualizes the results of *Dual-TF* for the five anomaly categories [32] in terms of t-anomaly and f-anomaly scores (second row) and then final anomaly scores (third row) in the TODS dataset. For point-wise anomalies (first and second columns), t-anomaly scores increase sharply at the actual locations. For pattern-wise anomalies (third, fourth, and fifth columns), f-anomaly scores maintain high values *throughout the entire anomaly interval*, whereas t-anomaly scores jump only at a few timestamps. As a result, each domain plays a distinct role and both point-wise and pattern-wise anomalies are precisely detected by the final scores.

## 4.5 Window Length Sensitivity

Because the window lengths, $w^{outer} = \rho \cdot w^{inner}$ and $w^{inner}$, are the most crucial hyperparameters in *Dual-TF*, we examine the effect of

varying these values on the detection accuracy in the TODS dataset. Figure 7(a) demonstrates the change in the $F_1$ score when $w^{inner}$ varies by $[0.2, 0.4, 1.0, 1.2, 1.6, 2.0]$ times the value determined in Section 3.2 while $\rho$ remains constant at 2. The proposed value for $w^{inner}$ clearly achieves the highest accuracy. Figure 7(b) shows the change in the $F_1$ score when $\rho$ varies within $[1, 2, 3, 4, 5]$ while maintaining the proposed value for $w^{inner}$. The highest accuracy is achieved when $\rho$ is 2 or 3, and we choose a smaller one to reduce the number of inner windows for efficiency.

**More Results in Supplementary Material:** Appendix E reports (i) the detection accuracy on the comprehensive UCR datasets, (ii) the effect of different backbone networks (e.g., autoencoders and conventional Transformers) on *Dual-TF*, (iii) the detection accuracy of all methods in other evaluation metrics, and (iv) the training and inference efficiency of *Dual-TF*. Furthermore, Appendix F shows the additional visualizations similar to Figure 6.

## 5 Conclusion

We define the concept of the *time-frequency granularity discrepancy* and formulate the problem in exploiting both the time and frequency domains for TSAD. To resolve this discrepancy, we employ the *NS-windowing* scheme to generate anomaly scores at the data-point granularity for both domains. The proposed framework is general and applicable to any sliding window-based TSAD method. This framework is implemented with *Dual-TF* on top of parallel Transformer architectures. Quantitative and qualitative evidence demonstrates the superiority of *Dual-TF*, especially in identifying pattern-wise anomalies with pinpoint accuracy. Overall, we believe that our work paves the way for a new approach to combining the time and frequency domains in time-series data analysis.

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

# A More Explanations on Time and Frequency Domains

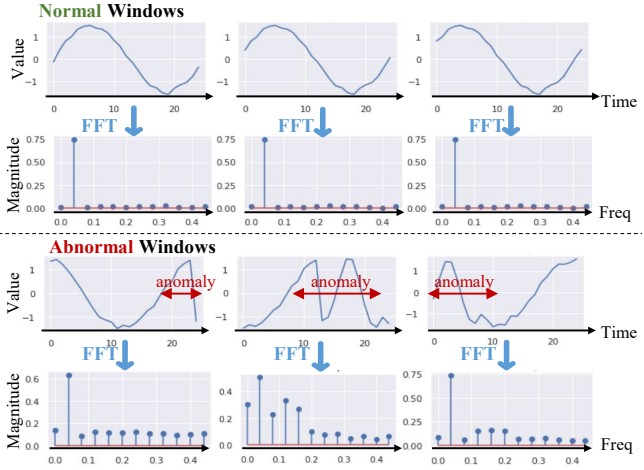

**Figure 8: Visualization for normal and abnormal windows in the time and frequency domains.**

Historically, point and collective anomalies were used as general criteria for any type of data, including images, videos, and time series [14, 40]. A point anomaly is an unanticipated occurrence at particular time points. A collective anomaly is defined as a collection of relational data instances that are anomalous in comparison to the entire data set. These anomalies can be further categorized as either global or contextual. While global anomalies are the points or groups that deviate significantly from the entirety of the data, contextual anomalies are the points or groups that deviate from their associated context, which is defined as the adjacent time points within specific ranges. On the basis of the above existing general taxonomy of anomalies, time-series anomaly detection algorithms have been developed to improve performance. Nevertheless, the collective anomaly of conventional taxonomy disregards the time-series temporal structure, such as trend and seasonal data. Even if two anomalous subsequences occurred for different reasons, such as an unusual shapelet and an extremely high frequency, they would be recognized as the same collective anomalies. In order to resolve the ambiguity of the current collective anomaly criterion, a new behavior-driven taxonomy [32] was proposed for time-series anomaly detection.

The behavior-driven taxonomy for time-series anomalies categorizes the anomalies as point-wise or pattern-wise. Pattern-wise anomalies are further divided based on the primary cause of anomaly occurrence into shapelet, seasonality, and trend anomalies. First, shapelet anomalies are the subsequences that deviate from typical shapelets. Dissimilarity measures, such as dynamic time warping, are often used to determine the degree of shapelet dissimilarity. Second, seasonal anomalies are the subsequences with a seasonality that is distinct from the overall seasonality. It is possible to specify the degree of seasonal variation in the *frequency* domain. Third, trend anomalies are the instances in a time series that significantly alter its trend, causing a long-term shift in its mean.

As shown in Figure 8, the anomalies that distinguish between normal and abnormal windows exhibit more prominent dissimilarities in the frequency domain compared to the time domain. Furthermore, we observe that expansion of the pattern-wise anomaly within the time window leads to a greater change in the frequency distribution within the frequency domain. That is, the frequency domain is more sensitive to changes in the underlying patterns of the time-series data. This noteworthy finding has served as the motivation for our proposed *Dual-TF*. We firmly believe that this discovery holds significance as it suggests that analyzing the frequency domain may offer a more effective approach for detecting pattern-wise anomalies than analyzing the time domain alone.

# B Details of the Window Length Selection

For a time series $X \in \mathbb{R}^{n \times d}$, the discrete Fourier transform (DFT) for the $d'$-th dimension ($d' \in [1, d]$) of $X$ is formulated as

$$X_k^{d'} = \sum_{t=0}^{n-1} x_t^{d'} \, e^{-\frac{i2\pi}{n}kt}, \tag{12}$$

where

$$x_t^{d'} = \frac{1}{n} \sum_{k=0}^{n-1} X_k^{d'} \cdot e^{-\frac{i2\pi}{n}kt}, \tag{13}$$

and $k$ is an integer ranging from 0 to $n-1$.

The radix-2 decimation rearranges the DFT of the function $x_t$ into two parts: a sum over the even-numbered indices $t = 2m$ and a sum over the odd-number indices $t = 2m + 1$,

$$X_k^{d'} = \sum_{m=0}^{n/2-1} x_{2m}^{d'} e^{-\frac{i2\pi}{n}k(2m)} + \sum_{m=0}^{n/2-1} x_{2m+1}^{d'} e^{-\frac{i2\pi}{n}k(2m+1)}. \tag{14}$$

Denote the DFT of the even-indexed inputs $x_{2m}^{d'}$ by $E_k^{d'}$ and the DFT of the odd-indexed inputs $x_{2m+1}^{d'}$ by $O_k^{d'}$, and we obtain

$$
\begin{aligned}
X_k^{d'} &= \sum_{m=0}^{n/2-1} x_{2m}^{d'} e^{-\frac{i2\pi}{n/2}km} + e^{-\frac{i2\pi}{n}k} \sum_{m=0}^{n/2-1} x_{2m+1}^{d'} e^{-\frac{i2\pi}{n/2}km} \\
&= E_k^{d'} + e^{-\frac{i2\pi}{n}k} O_k^{d'} \quad \text{for } k = 0, \dots, \frac{n}{2} - 1.
\end{aligned}
\tag{15}
$$

Due to the periodicity of the complex exponential, $X_{k+\frac{n}{2}}^{d'}$ can be also represented as

$$
\begin{aligned}
X_{k+\frac{n}{2}}^{d'} &= \sum_{m=0}^{n/2-1} x_{2m}^{d'} e^{-\frac{i2\pi}{n/2}m(k+\frac{n}{2})} \\
&\quad + e^{-\frac{i2\pi}{n}(k+\frac{n}{2})} \sum_{m=0}^{n/2-1} x_{2m+1}^{d'} e^{-\frac{i2\pi}{n/2}m(k+\frac{n}{2})} \\
&= \sum_{m=0}^{n/2-1} x_{2m}^{d'} e^{-\frac{i2\pi}{n/2}mk} e^{-i2\pi m} \\
&\quad + e^{-\frac{i2\pi}{n}k} e^{-i\pi} \sum_{m=0}^{n/2-1} x_{2m+1}^{d'} e^{-\frac{i2\pi}{n/2}mk} e^{-i2\pi m} \\
&= \sum_{m=0}^{n/2-1} x_{2m}^{d'} e^{-\frac{i2\pi}{n/2}mk} - e^{-\frac{i2\pi}{n}k} \sum_{m=0}^{n/2-1} x_{2m+1}^{d'} e^{-\frac{i2\pi}{n/2}mk} \\
&= E_k^{d'} - e^{-\frac{i2\pi}{n}k} O_k^{d'}.
\end{aligned}
\tag{16}
$$

Rewrite $X_k^{d'}$ and $X_{k+\frac{n}{2}}^{d'}$ to

$$X_k^{d'} = E_k^{d'} + e^{-\frac{i2\pi}{n}k} O_k^{d'}$$
$$X_{k+\frac{n}{2}}^{d'} = E_k^{d'} - e^{-\frac{i2\pi}{n}k} O_k^{d'}. \quad (17)$$

Note that the final outputs are obtained by a $+/-$ combination of $E_k^{d'}$ and $O_k^{d'} e^{-\frac{i2\pi}{n}k}$. Therefore, we can use only one of the two parts to get information about the magnitude through the absolute value (because both have the same magnitude if the absolute value is taken). We can define the number of sampling rate (sampling frequency) per time unit as $f$ obtained from a continuous signal. The frequency range is expressed as $f = \frac{k}{n}$ for $k = 0, \ldots, \frac{n}{2} - 1$, where $n$ is the time period.

Among $d$ dimensions, the value with the smallest frequency is set as the most dominant frequency. The reason for choosing the minimum frequency is the allowance for the inclusion of other dominant frequencies in other different dimensions within the determined inner window. We define the maximum magnitude index, which means the most dominant frequency, $v_{\text{major}}$,

$$v_{\text{major}} = \min(\arg\max(|X_f^{d'}|) : d' \in [1, d]) \quad (18)$$
$$\text{for } f = 0, \ldots, \frac{n-2}{2n}.$$

Finally, the inner window length can be determined as

$$w^{inner} = \lceil \frac{1}{v_{\text{major}}} \rceil. \quad (19)$$

## C Proof of Theorem 3.1

The proof can be done using Parseval's theorem [27]. By our definition of the uncertainty in each domain, $\mathcal{U}^{time}(w^{inner})$ monotonically decreases as $w^{inner}$ decreases; $\mathcal{U}^{freq}(w^{inner})$ converges to a zero when $w^{inner} \geq \lceil \frac{1}{v_{major}} \rceil$, but it monotonically increases as $w^{inner}$ decreases when $w^{inner} < \lceil \frac{1}{v_{major}} \rceil$.

Specifically, in the process of interpreting the results obtained through the NS-windowing and the frequency reconstruction by $\Theta^{freq}$, there is uncertainty between two different domains. From the perspective of the time domain, the smaller $w^{inner}$ facilitates the identification of abnormal time points. Conversely, as $w^{inner}$ increases, distinguishing time points within a window in the frequency domain becomes more challenging. On the other hand, from the perspective of the frequency domain, the longer $w^{inner}$ leads to a decrease in frequency variance according to the uncertainty principle, and thus, frequency becomes concentrated. However, a decrease in $w^{inner}$ makes it difficult to accurately determine the precise frequency after the Fourier transform.

We formulate the inherent trade-off associated with dual-domain information loss. Assume that every time series can be represented as a single periodic function with a dominant frequency that affects the pattern most. Then, the time series can be simply expressed as $x_t = \cos(2\pi v_{major} \frac{t}{w}) + \epsilon_t$, where $t = 0, \ldots, w-1$ with the dominant frequency $v_{major}$ and a noise $\epsilon_t \sim N(0, w)$.

Here, $\mathcal{U}(w)$ in Eq. (1) is a function of $w$, which represents the dual-domain information loss. $\mathcal{U}^{time}(w)$ denotes the uncertainty in the time domain, which monotonically decreases as $w$ decreases as a linear function of $w$ ($w \geq 1, w \in \mathbb{N}$), so we define it as

$$\mathcal{U}^{time}(w) = w - 1. \quad (20)$$

The choice of a linear relationship is specific to the Gaussian function and the Fourier transform. The Gaussian function has the unique property that its Fourier transform is also a Gaussian function, and this symmetry leads to the linear relationship between the standard deviations (or uncertainties) in time and frequency. While other functions, such as square, cubic, or exponential functions, can be used to model specific types of uncertainty, they would not lead to the same fundamental relationship as the Gaussian function does in the context of the Gabor limit. The Gaussian function is crucial due to its role in signal processing and its mathematical properties that align with both time and frequency domains. Related works [8, 15] provide more detailed insights into the mathematical reasoning behind the linear relationship in the Gabor uncertainty principle.

$\mathcal{U}^{freq}(w)$ denotes the uncertainty in the frequency domain. $\widehat{F}_v$ denotes a magnitude of frequency $v$ after the Fourier transform (or DFT) of $x_t$, where $v = 0, \ldots, w - 1$ ($v = 0, \ldots, \frac{w}{2} - 1$ for DFT). $\widehat{F}_v$ is formulated as $\widehat{F}_v = F_v + E_v$, where $F_v$ is the clean (unperturbed) Fourier transform and $E_v$ is the Fourier transform of the noise $\epsilon_t$. $\mathcal{U}^{freq}(w)$ can be defined as the sum of the standard deviation of the Fourier transformed $\widehat{F}_v$, i.e., $\sigma(E_v)$ and the function of $F_1$ scores (inversely related to the anomaly scores) $f(w)$ with varying the length of $w$ in Figure 7, where $w = 1/v$ (by the definition of a period in terms of frequency).

We approximate the anomaly score function $f$ by fitting the function $aw - \log(bw)$ to Figure 7. To justify the fitting approach, let us conduct a T-test on a custom fitting function $f$ with two other possible functions, i.e., linear function $-ax + b$ and rational function $\frac{a}{x} + b$. We can fit the custom function $f$ using *spicy curve_fit* library and calculate the standard error of the parameters from the covariance. The each null hypothesis of $a$ and $b$ values is

$$H_0 : \mu(\Delta_\phi^{custom}) = \mu(\Delta_\phi^{linear}),$$
$$H_0 : \mu(\Delta_\phi^{custom}) = \mu(\Delta_\phi^{rational}), \quad (21)$$

where $\Delta_\phi^{custom}$ is the standard error of parameter $\phi \in \{a, b\}$ in the custom function (identical meaning in both linear and rational functions). The maximum p-value is 0.029 among all pairs with a significance level of 0.05. Therefore we reject the null hypothesis for every pair with the custom fitting function, meaning that the standard deviations of parameters are less than the other two compared fitting functions. Additionally, by fitting the $f$ using the data in Figure 7, the value of $a$ becomes $v_{major} - 1$, and the value of $b$ becomes 1 after fitting with the R-square value of 0.695–0.972, which is much higher than the other compared functions in TODS benchmark datasets.

By Parseval's theorem [27], the sum of the square of a function is equal to the sum of the square of its transform, $\sum |E_v|^2 = \frac{1}{w} \sum |\epsilon_t|^2 = \sigma(\epsilon_t)^2$. Therefore,

$$\sigma(E_v) = \sqrt{\frac{1}{w} \sum |E_v|^2} = \sqrt{\frac{\sigma(\epsilon_t)^2}{w}} = 1. \quad (22)$$

Finally,

$$\mathcal{U}^{freq}(w) = f(v) + \sigma(E_v)$$
$$= (v_{major} - 1)w - \log w + 1. \quad (23)$$

---

**Algorithm 1** Algorithm of *Dual-TF* for Training and Inference Phases

---

INPUT: $\mathcal{X} \in \mathbb{R}^{n \times d}$: training set, $\mathcal{X}' \in \mathbb{R}^{n' \times d}$: test set
OUTPUT: $y = \{y_1, y_2, \ldots, y_{n'}\}$

1: $\mathcal{OW} = \text{Windowing}(\mathcal{X})$      $\mathcal{OW} \in \mathbb{R}^{(n - w^{outer} + 1) \times w^{outer} \times d}$
2: $\mathcal{IW} = \text{NS-Windowing}(\mathcal{OW})$     $\mathcal{IW} \in \mathbb{R}^{(n - w^{outer} + 1) \times (w^{outer} - w^{inner} + 1) \times w^{inner} \times d}$
3: $\Theta^{time}, \Theta^{freq} \leftarrow \text{initialize weights}$
4: /* Training phase */
5: **for** $epoch = 1$ **to** $epochs$ **do**
6:     $\widehat{\mathcal{OW}} = \Theta^{time}(\mathcal{OW})$     Eq. (2)
7:     $\mathcal{L}^{time}(\mathcal{OW}) \leftarrow RecLoss^{time} - \lambda \cdot AssDis^{time}$     Eq. (4), Eq. (5)
8:     $\widehat{\mathcal{IW}} = \Theta^{freq}(\mathcal{IW})$     Eq. (2)
9:     $\mathcal{L}^{freq}(\mathcal{IW}) \leftarrow RecLoss^{freq} - \lambda \cdot AssDis^{freq}$     Eq. (6), Eq. (7)
10:    $\Theta^{time}, \Theta^{freq} \leftarrow \text{update weights using } \mathcal{L}^{time} \text{ and } \mathcal{L}^{freq}, \text{ respectively}$
11: **end for**
12: /* Inference phase */
13: $\widehat{\mathcal{OW'}} = \Theta^{time*}(\mathcal{OW'})$     $\mathcal{OW'} \in \mathbb{R}^{(n' - w^{outer} + 1) \times w^{outer} \times d}$
14: $\widehat{\mathcal{IW'}} = \Theta^{freq*}(\mathcal{IW'})$     $\mathcal{IW'} \in \mathbb{R}^{(n' - w^{outer} + 1) \times (w^{outer} - w^{inner} + 1) \times w^{inner} \times d}$
15: **for** $t = 1$ **to** $n'$ **do**
16:    $\mathcal{A}^{time}(\mathbf{x}_t) \leftarrow \text{Softmax}(-AssDis^{time}) \odot RecLoss^{time}$     Eq. (8)
17:    $\mathcal{A}^{freq}(\mathbf{x}_t) \leftarrow \text{Softmax}(-AssDis^{freq}) \odot RecLoss^{freq}$     Eq. (9)
18:    $\mathcal{A}^{total}(\mathbf{x}_t) = \mathcal{A}^{time}(\mathbf{x}_t) + \mathcal{A}^{freq}(\mathbf{x}_t)$     Eq. (11)
19:    **if** $\mathcal{A}^{total} > \delta$ **then**
20:      $y_t \leftarrow 1$ /* identify as an anomalous value */
21:    **else**
22:      $y_t \leftarrow 0$ /* identify as a normal value */
23:    **end if**
24: **end for**
25: **return** $y : \{y_1, y_2, \ldots, y_{n'}\}$

---

The uncertainty in the two domains is

$$\mathcal{U}(w) = \mathcal{U}^{time}(w) + \mathcal{U}^{freq}(w)$$
$$= w - 1 + (v_{major} - 1)w - \log w + 1. \quad (24)$$

The length that minimizes Eq. (24) (or Eq. (1)) can be obtained by solving the differential equation for $\mathcal{U}(w)$ with respect to $w$. Taking the derivative of $\mathcal{U}(w)$ with respect to $w$,

$$\frac{\partial \mathcal{U}(w)}{\partial w} = \frac{\partial(w - 1 + (v_{major} - 1)w - \log w + 1)}{\partial w}$$
$$= 1 + v_{major} - 1 - \frac{1}{w} \quad (25)$$
$$= v_{major} - \frac{1}{w} = 0.$$

We now find the solution with $w = \frac{1}{v_{major}}$, and the length of the inner window is determined as $w^{inner} = \lceil \frac{1}{v_{major}} \rceil$ that is consistent with Eq. (19). This completes the proof. □

# D Details of the Experiment Settings

## D.1 Pseudo-code of *Dual-TF*

We adopt Anomaly Transformer [55] as the underlying backbone network for the reconstructor. Both the time and frequency reconstructors employed in the *Dual-TF* operate on the same fundamental mechanism as Anomaly Transformer, differing only in input and output dimensions. We provide the pseudo-code implementation of the training and inference phases for the *Dual-TF* in Algorithm 1. In summary, the pseudo-code describes the entire flow of the *Dual-TF* algorithm. Each step involving the calculation of the loss or anomaly score is computed using the equations defined in Section 3. Upon inputting a time series, the *Dual-TF* generates a binary outcome for each time point, indicating whether it is classified as an anomaly or not.

## D.2 Datasets

We evaluate anomaly detection performance on a total of 30 datasets derived from five benchmarks. A detailed description of the four publicly available datasets is provided as follows.

**ASD** [34] represents an application server benchmark consisting of 45-day-long multivariate time series. The benchmark comprises 12 entities obtained from different servers, with each entity characterized by 19 metrics that reflect the server's status. These metrics include CPU-related parameters, memory-related parameters, network metrics, and virtual machine metrics. The ASD benchmark is publicly available under the MIT license at https://github.com/zhhlee/InterFusion/tree/main/data.

**ECG** [28] (Electrocardiogram) represents time series of the electrical potential variation between two points on the body's surface, primarily originating from the rhythmic contractions of the heart. The benchmark contains 9 sub-datasets. Detailed statistical information regarding the ECG sub-datasets can be found in Table 6.

**Table 5: Experiment environments for all algorithms.**

| Env. | OmniAnomaly | LOF | ISF | OCSVM | VAE | MS-RNN | RANSynCoders | TranAD | Anomaly Transformer | *Dual-TF* |
|---|---|---|---|---|---|---|---|---|---|---|
| Library | Tensorflow 1.12.0 | Scikit-Learn 1.2.1 | | | Tensorflow 2.5.0 | | | | PyTorch 1.13.1 | |
| CPU | Intel(R) Xeon(R) Silver 4116 CPU @ 2.10GHz | Intel(R) Xeon(R) Gold 6226R CPU @ 2.90GHz | | | | | | | | |
| GPU | NVIDIA GeForce RTX 2080 Ti 11GB with CUDA Version 11 | NVIDIA GeForce RTX 3090 24GB with CUDA Version 11 | | | | | | | | |

**Table 6: Detailed statistics of ECG sub-datasets.**

| Sub-Datasets | Dim. | # Train | # Test | # Pattern Anomaly | Anomaly Percentage (%) |
|---|---|---|---|---|---|
| chfdb_chf01_275 | | 1,833 | 1,841 | 269 | 14.61 |
| chfdb_chf13_45590 | | 2,439 | 1,287 | 159 | 12.35 |
| chfdbchf15 | | 10,863 | 3,348 | 149 | 4.45 |
| lstdb_20221_43 | | 2,610 | 1,121 | 129 | 11.51 |
| lstdb_20321_240 | 2 | 2,011 | 1,447 | 139 | 9.61 |
| mitdb_100_180 | | 2,943 | 2,255 | 189 | 8.38 |
| qtdbsel102 | | 34,735 | 9,882 | 199 | 2.01 |
| stdb_308_0 | | 2,373 | 2,721 | 259 | 9.52 |
| xmitdb_x108_0 | | 3,152 | 1,756 | 379 | 21.58 |

The ECG benchmark is publicly accessible at https://www.cs.ucr.edu/~eamonn/discords/ECG_data.zip.

**PSM** [2] represents a benchmark comprising a single entity derived from multiple application server nodes at eBay. It encompasses 26 features, publicly accessible under the CC BY 4.0 license at https://github.com/eBay/RANSynCoders/tree/main/data. Similar to ASD, these features describe server machine metrics such as CPU utilization and memory usage. In the PSM benchmark, the training set spans 13 weeks, followed by 8 weeks for testing.

**TODS** [32] serves as a synthetic benchmark and data generator designed for time-series anomaly detection. It is publicly available at https://github.com/datamllab/tods/tree/benchmark/benchmark/synthetic/Generator. The benchmark comprises five distinct anomaly scenarios for time-series data, which are classified based on a behavior-driven taxonomy. These scenarios include point-global, pattern-contextual, pattern-shapelet, pattern-seasonal, and pattern-trend anomalies. The data generator provided by TODS allows the generating of 5 individual univariate time series, with each series corresponding to a distinct anomaly type. In order to ensure a fair comparison, we employ the available source code without any alterations, except for adjusting the length parameter to generate longer time series, as demonstrated below.

```
1  # Source: https://github.com/datamllab/tods
2  # DataGenerator: "UnivariateDataGenerator" from
       univariate_generator.py.
3  BEHAVIOR_CONFIG = {"freq": 0.04, "coef": 1.5, "offset": 0.0, "
       noise_amp": 0.05}
4  BASE = [0.145, 0.128, 0.094, 0.077, 0.111, 0.145, 0.179, 0.214,
       0.214]
5
6  # Training set
7  normal = DataGenerator(stream_length=20000,
8                         behavior=sine,
9                         behavior_config=BEHAVIOR_CONFIG)
10 normal.generate_timeseries()
11
12 # Test set
```

```
13 test = DataGenerator(stream_length=5000,
14                      behavior=sine,
15                      behavior_config=BEHAVIOR_CONFIG)
16
17 # Point - global anomaly
18 if anomaly_type=='point_global':
19     test.point_global_outliers(ratio=0.05, factor=3.5, radius=5)
20 elif anomaly_type=='point_contextual':
21     test.point_contextual_outliers(ratio=0.05, factor=2.5, radius
       =5)
22 elif anomaly_type=='pattern_shaplet':
23     test.collective_global_outliers(ratio=0.05, radius=5, option='
       square', coef=1.5, noise_amp=0.03, level=20, freq=0.04, base=
       BASE, offset=0.0)
24 elif anomaly_type=='pattern_seasonal':
25     test.collective_seasonal_outliers(ratio=0.05, factor=3, radius
       =5)
26 elif anomaly_type=='pattern_trend':
27     test.collective_trend_outliers(ratio=0.05, factor=0.5, radius
       =5)
```

**Listing 1: The command used for generating the TODS benchmark datasets.**

### D.3 Baselines

**LOF** is an unsupervised outlier detector that measures the local deviation of the density of a given sample to its neighbors. **ISF** is a well-known anomaly detection algorithm that works on the principle of isolating anomalies using tree-based structures. **OCSVM** is an unsupervised outlier detection algorithm based on the SVM that maximizes the margin between the origin and the normality and defines the decision boundary as the hyper-plane that determines the margin. **VAE** is a simple neural architecture that uses the symmetrical encoder and decoder network for anomaly detection. Anomaly scores are the differences between the inputs and reconstructed outputs.

The experiments are conducted also using the following state-of-the-art methods: Modified-RNN (**MS-RNN**), a modified version of an anomaly detector that exploits sparsely-connected recurrent neural networks (RNNs) and an ensemble of sequence-to-sequence autoencoders (AEs) for multi-resolution learning; **OmniAnomaly**, a LSTM-based variational autoencoder (**VAE**) that captures complex temporal dependency between multivariate time series and maps observations to stochastic variables; **RANSynCoders**, a model that utilizes pre-trained AEs to extract primary frequencies across the signals on the latent representation for synchronizing time series; **TranAD**, a Transformer-based model that uses attention-based sequence encoders to perform inference with broader temporal trends in time series, with the focus on score-based self-conditioning for robust multi-modal feature extraction and adversarial training for stability; **Anomaly Transformer**, a reconstructive approach that

**Table 7: Creator-suggested binary accuracy for 250 UCR datasets. "C" is an abbreviation for a correct detection, and "I" for an incorrect detection.**

| idx | 1 | 2 | 3 | 4 | 5 | 6 | 7 | 8 | 9 | 10 | 11 | 12 | 13 | 14 | 15 | 16 | 17 | 18 | 19 | 20 | 21 | 22 | 23 | 24 | 25 |
|-----|---|---|---|---|---|---|---|---|---|----|----|----|----|----|----|----|----|----|----|----|----|----|----|----|----|
| 1 | C | I | I | C | C | C | C | C | C | C | C | I | C | I | I | C | C | C | C | I | C | C | C | C | C |
| 2 | C | C | C | C | C | C | C | C | C | C | C | C | I | C | C | I | C | C | C | I | C | C | C | C | C |
| 3 | C | C | C | C | I | I | I | I | C | C | I | I | I | C | C | C | C | C | C | C | C | C | C | I | I |
| 4 | C | C | I | I | I | I | I | C | C | C | I | C | I | I | C | I | I | C | C | I | C | I | C | C | C |
| 5 | C | C | C | I | C | I | C | I | C | I | C | C | C | C | C | C | C | C | C | C | C | C | C | C | C |
| 6 | C | C | I | C | C | C | I | C | C | C | C | C | C | C | C | C | C | C | C | C | C | C | C | C | C |
| 7 | C | C | I | C | C | C | C | C | C | C | C | C | C | I | I | I | C | C | C | I | I | C | I | C | C |
| 8 | C | C | C | C | I | C | I | I | C | C | I | I | I | I | I | C | C | C | I | C | C | I | I | I | C |
| 9 | I | I | I | C | I | I | I | C | C | I | I | I | C | C | I | C | I | C | C | I | C | C | I | C | I |
| 10 | C | I | I | I | C | I | I | C | I | C | I | C | C | I | I | I | I | I | I | I | I | I | C | C | I |

**Table 8: Comparison with previous methods for 250 UCR datasets. The accuracy of the existing methods is taken from Table 7 of Lu et al. [37]; the number of correct predictions is simply derived by multiplying the accuracy by 250.**

| Method | # Correct | Accuracy |
|--------|-----------|----------|
| USAD [5] | 69.0 | 0.276 |
| LSTM-VAE [43] | 49.5 | 0.198 |
| AE [4] | 59.0 | 0.236 |
| NORMA [9] | 118.5 | 0.474 |
| SCRIMP [65] | 104.0 | 0.416 |
| DAMP (out-of-the-box) [36, 37] | 128.0 | 0.512 |
| DAMP (sharpened data) [36, 37] | 158.0 | 0.632 |
| *Dual-TF* | **164.0** | **0.656** |

combines series and prior association to make anomalies distinctive; and **TFAD**, a time-frequency analysis-based anomaly detection model that utilizes both time and frequency domains, with time series decomposition and data augmentation mechanisms to enhance performance and interpretability.

## D.4 Experiment Environments

Table 5 shows the experiment hardware environments for all algorithms. Only OmniAnomaly was run on a different environment due to its incompatibility between the NVIDIA cuDNN and Tensorflow libraries. Despite the presence of multiple GPUs within the server infrastructure, we adopt a singular GPU for conducting experiments on each benchmark and algorithm. All experiments are conducted on a server equipped with an NVIDIA RTX 3090Ti.

## D.5 Model Hyperparameter Settings

The ISF, LOF, and OCSVM algorithms are implemented using the Scikit-Learn library, while the remaining methods are configured using open-source code obtained from each URL. The hyperparameters for the baseline methods are set as follows.

- **ISF**: The number of tree is selected from $\{25, 100\}$.
- **LOF**: The number of neighbors is selected from $\{1, 3, 5, 12\}$.
- **OCSVM**: The RBF kernel is used. The inverse length parameter $\gamma$ is selected from $\{10^{-4}, 10^{-3}, 10^{-2}, 10^{-1}, 0.5\}$.
- **VAE**[2]: The LSTM layers are used as both the encoder and decoder. The number of hidden units in the encoder and decoder are set to 64 and 32, respectively.
- **MS-RNN**[3]: The GRU layers are employed as the encoder, and the decoder consists of a skip-GRU structure with a reverse chronological order in the time series.
- **OmniAnomaly**[4]: The GRU and dense layers have 500 units. The standard deviation layer has an $\epsilon$ value of $10^{-4}$. The dimension of the latent variable $z$-space is fixed at 3.
- **RANSynCoders**[5]: The number of hidden layers in each corresponding decoder is increased because the output dimension is

---

[2]https://github.com/lin-shuyu/VAE-LSTM-for-anomaly-detection
[3]https://github.com/tungk/OED
[4]https://github.com/NetManAIOps/OmniAnomaly
[5]https://github.com/eBay/RANSynCoders

at least 3 times larger than the encoder input size. The values of $S$, $N$, and the bootstrap sample size are selected as one-third of the input dimension, rounded to the nearest multiple of 5. The number of latent dimensions is selected as $0.5N-1$. The value of $\delta$ is set to 0.05 for system data with Gaussian outliers and 0.1 for business data without Gaussian outliers.

- **TranAD**[6]: The number of layers in the Transformer encoders is set to 1. The number of layers in the feed-forward unit of the encoders is 2. The number of the hidden units in the encoder layers is set to 64, and the dropout in the encoders is set to 0.1.
- **TFAD**[7]: The kernel size for the temporal convolutional network (TCN) is set to 7. The number of TCN layers is 3. The dimension of the embedding representation is set to 150. The distance metric is the L2 norm. The classifier threshold is set to 0.5, and the mixup rate is set to 2.
- **Anomaly Transformer**[8]: The number of layers is 3, the channel number of hidden states $d_{model}$ is 512, and the number of heads $h$ is 8. The loss function hyperparameter $\lambda$ for balancing two parts is set as 3. These hyperparameter settings are shared with *Dual-TF*.

---

[6]https://github.com/imperial-qore/TranAD
[7]https://github.com/DAMO-DI-ML/CIKM22-TFAD
[8]https://github.com/thuml/Anomaly-Transformer

**Table 9: Accuracy comparison with different backbones in the best point-wise $F_1$ score.**

| Methods | TODS (Point) | | TODS (Pattern) | | |
| --- | --- | --- | --- | --- | --- |
| | Gloabl | Contextual | Shaplet | Seasonal | Trend |
| Vanilla AE | 0.839 | 0.553 | 0.165 | 0.812 | 0.180 |
| *Dual-TF*@AE | 0.956 | 0.661 | 0.459 | 0.901 | 0.227 |
| *Dual-TF* | **0.968** | **0.943** | **0.741** | **0.925** | **0.476** |

## E  Additional Experiment Results

### E.1  Performance on the UCR Benchmark

**UCR Benchmark** [9] [29] is created for KDD Cup 2021 and designed to mitigate previous benchmark problems. All 250 datasets are composed of univariate time series covering various real-world scenarios. These datasets include a wide range of domains such as cardiology, industry, medicine, zoology, weather, and human behavior. The datasets in the benchmark have varying lengths, ranging from 6,684 to 900,000 data points, and have been divided into separate training and test sets.

We use the accuracy metric that was suggested by the dataset creators. In summary, each of the 250 datasets contains a single anomaly, and the task of an anomaly detector is to predict its location. Let the length of the anomaly be $L$. If the prediction is within plus or minus $L$ data points of the anomaly's ground truth location, it is considered a correct prediction. The results in Table 7 show the binary indices for each dataset, and the *final accuracy* indicates the ratio of the number of correct predictions to the total number of datasets 250. A table consisting of 10-row indices and 25-column indices represents the dataset number in sequential order. In detail, each cell corresponds to the dataset number within the table, which stands for $(25 \times (\text{idx}^{\text{row}} - 1) + \text{idx}^{\text{column}})$-th dataset. Table 8 shows the number of correct predictions and the final accuracy for the 250 datasets with existing methods. The final accuracy achieved by *Dual-TF* on the UCR benchmark is **0.656**, surpassing the previously-reported highest accuracy of 0.632 [37].

### E.2  Effect of Different Backbone Networks

To show that *Dual-TF* can be built on top of any reconstruction-based networks, we replace the Anomaly Transformer backbone with a GRU autoencoder backbone.

Table 9 reports the best point-wise $F_1$ score when the autoencoder backbone is employed in *Dual-TF* (denoted as *Dual-TF*@AE). Following the performance of the backbone itself, the result indicates that the original *Dual-TF* with Anomaly Transformer provides a more precise detection. If a new backbone that surpasses Anomaly Transformer is developed, it is expected that the performance of *Dual-TF* can be enhanced simply by adopting the new backbone due to its high adaptability.

### E.3  Evaluation with the Point-Adjusted Metric

The widely-used point-adjusted (PA) metric has overestimation issues [49, 54, 55], as a window is considered correct if both a predicted anomaly and the true anomaly occur just within the same window. Despite the fact that the PA metric is inappropriate for our work, we report the results using the PA metric so that they can be compared to the findings of other studies.

Table 10 shows the performance comparison between TSAD methods with another TSAD metric—the best *point-adjusted (PA)* $F_1$ score. This table is useful for comparing existing literature employing the same metric. Overall, *Dual-TF* is shown to outperform other TSAD methods also in the PA metric. Note that some errors of the detected locations are accounted for in the PA metric, which diminishes the advantages of *Dual-TF* over the other methods in

---

[9] https://compete.hexagon-ml.com/practice/competition/39/

comparison to Table 2, which uses the point-wise metric. While the *Dual-TF* did not achieve the highest score in the TODS (Pattern) Trend dataset, it attained the second-highest score, exhibiting only a marginal difference. The PA $F_1$ score improves by 12.6–42.3% on average compared with the other methods.

**Table 11: Comparison of computation time with Anomaly Transformer in real-world datasets.**

| Phase | Network | ASD | ECG | PSM | CompanyA |
| --- | --- | --- | --- | --- | --- |
| Train(sec/iter) | Anomaly Transformer | 0.043 | 0.040 | 0.044 | 0.065 |
| | *Dual-TF* | 0.115 | 0.091 | 0.117 | 0.106 |
| Inference(sec) | Anomaly Transformer | 17.370 | 85.684 | 279.058 | 41.777 |
| | *Dual-TF* | 35.938 | 194.123 | 803.952 | 95.829 |

### E.4  Computation Efficiency

As in Table 11, *Dual-TF* takes longer 1.4–2.7 times for tranining and 1.9–2.9 times for inference than Anomaly Transformer. This additional cost for the improved accuracy is reasonable, considering the *two* reconstructors and the NS-windows. Because the inference time per data point is only **6.18–9.15 ms**, we believe that *Dual-TF* can also be deployed in a real-time environment.

## F  Additional Visualization Results

Because time-series analysis is inherently a visual domain, we would like to provide more visualization of the anomaly scores. Figure 9 provides the visualization for other sequences, including the anomaly scores of Anomaly Transformer and TFAD (the second and the third best methods). Again, Figure 9 confirms that *Dual-TF* recognizes the pattern-wise anomalies very precisely by virtue of utilizing the two domains at the data-point granularity. On the other hand, the boundaries of the pattern-wise anomalies in TFAD are not very clearly identified due to the limitation of the window granularity. Anomaly Transformer partially detects pattern-wise anomalies. These visualizations (Figure 6 and Figure 9) along with the previous quantitative experiment results (Table 2, Table 3, and Table 10) demonstrate that t-anomaly and f-anomaly scores each play a distinct role and that their collaboration at the data-point granularity greatly improves the capability of precisely detecting pattern-wise anomalies.

**Table 10: Accuracy comparison between TSAD methods in terms of the best *point-adjusted (PA)* $F_1$ score with the highest scores highlighted in bold.**

| Methods | TODS (Point) | | TODS (Pattern) | | | ASD | ECG | PSM | CompanyA | Avg. ↑ | Rank ↓ |
| | Gloabl | Contextual | Shapelet | Seasonal | Trend | | | | | | |
| --- | --- | --- | --- | --- | --- | --- | --- | --- | --- | --- | --- |
| ISF | 0.973 (±0.000) | 0.553 (±0.000) | 0.165 (±0.000) | 0.162 (±0.000) | 0.293 (±0.000) | 0.339 (±0.000) | 0.262 (±0.000) | 0.498 (±0.000) | 0.198 (±0.000) | 0.383 (±0.000) | 11 |
| LOF | 0.947 (±0.000) | 0.553 (±0.000) | 0.123 (±0.000) | 0.127 (±0.000) | 0.147 (±0.000) | 0.487 (±0.000) | 0.346 (±0.000) | 0.587 (±0.000) | 0.480 (±0.000) | 0.422 (±0.000) | 9 |
| OCSVM | 0.990 (±0.000) | 0.553 (±0.000) | 0.165 (±0.000) | 0.162 (±0.000) | 0.188 (±0.000) | 0.374 (±0.000) | 0.277 (±0.000) | 0.474 (±0.000) | 0.342 (±0.000) | 0.392 (±0.000) | 10 |
| VAE | 0.959 (±0.027) | 0.855 (±0.120) | 0.721 (±0.106) | 0.912 (±0.060) | 0.409 (±0.183) | 0.413 (±0.029) | 0.282 (±0.008) | 0.477 (±0.042) | 0.296 (±0.030) | 0.592 (±0.040) | 5 |
| MS-RNN | 0.941 (±0.077) | 0.798 (±0.034) | 0.433 (±0.264) | 0.809 (±0.025) | 0.307 (±0.024) | 0.425 (±0.007) | 0.287 (±0.002) | 0.575 (±0.002) | 0.263 (±0.003) | 0.537 (±0.013) | 6 |
| OmniAnomaly | 0.964 (±0.035) | 0.856 (±0.003) | 0.388 (±0.182) | 0.279 (±0.137) | 0.377 (±0.186) | 0.238 (±0.155) | 0.284 (±0.111) | 0.552 (±0.002) | 0.383 (±0.000) | 0.480 (±0.047) | 8 |
| RANSynCoders | 0.934 (±0.099) | 0.921 (±0.117) | 0.507 (±0.196) | 0.459 (±0.178) | 0.512 (±0.165) | 0.384 (±0.227) | 0.305 (±0.018) | 0.601 (±0.003) | 0.200 (±0.042) | 0.536 (±0.062) | 7 |
| TranAD | 0.991 (±0.000) | 0.989 (±0.000) | 0.620 (±0.000) | 0.563 (±0.000) | **0.633** (±0.054) | 0.566 (±0.010) | 0.500 (±0.030) | 0.685 (±0.003) | 0.452 (±0.042) | 0.666 (±0.062) | 3 |
| TFAD | 0.960 (±0.003) | 0.937 (±0.020) | 0.736 (±0.009) | 0.859 (±0.014) | 0.561 (±0.050) | 0.517 (±0.004) | 0.453 (±0.039) | 0.672 (±0.067) | 0.416 (±0.026) | 0.679 (±0.016) | 2 |
| Anomaly Transformer | 0.942 (±0.014) | 0.944 (±0.001) | 0.420 (±0.276) | 0.603 (±0.250) | 0.485 (±0.017) | 0.585 (±0.012) | 0.547 (±0.070) | 0.957 (±0.009) | 0.391 (±0.022) | 0.653 (±0.056) | 4 |
| *Dual-TF* | **0.991** (±0.000) | **0.990** (±0.002) | **0.802** (±0.007) | **0.955** (±0.026) | 0.630 (±0.023) | **0.832** (±0.006) | **0.643** (±0.083) | **0.958** (±0.036) | **0.506** (±0.033) | **0.805** (±0.023) | 1 |

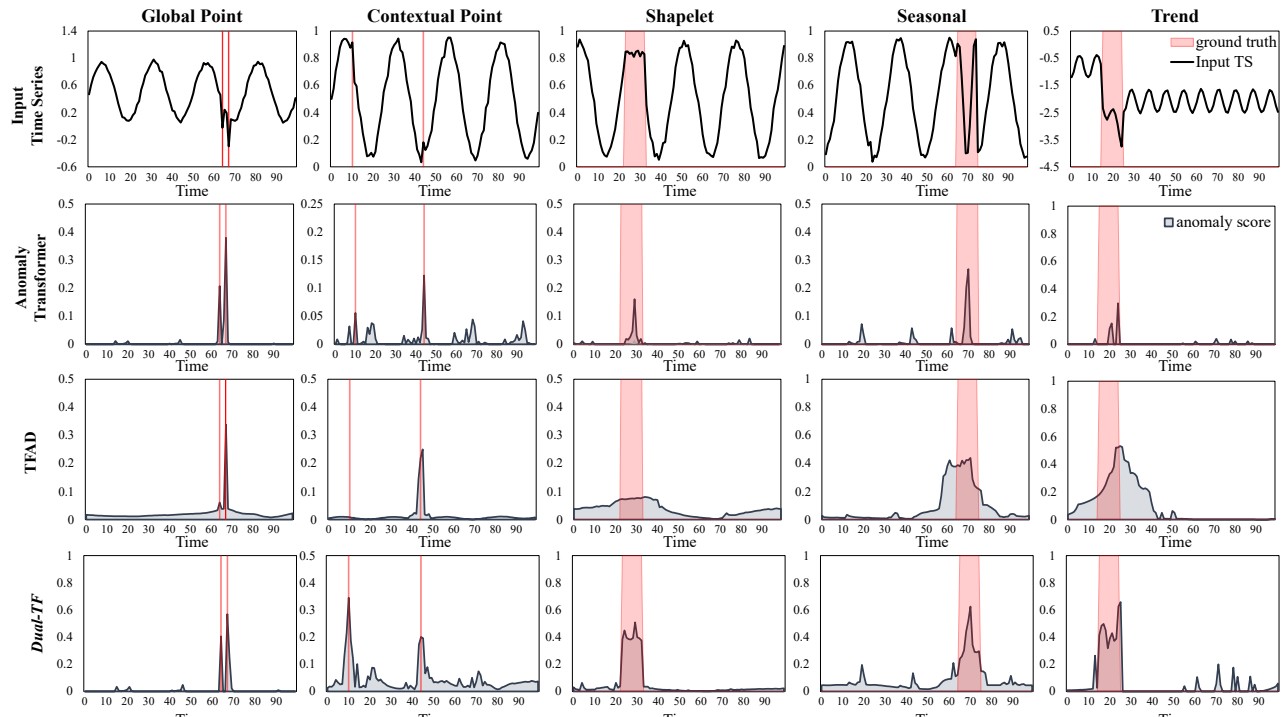

**Figure 9: Visualization of the anomaly scores from the top-3 methods for different categories of point- and pattern-wise anomalies using the TODS dataset.**

