# OpenReview forum: "Breaking the Time-Frequency Granularity Discrepancy in Time-Series Anomaly Detection"
_ACM.org/TheWebConf/2024/Conference — TheWebConf24_

### Official Review · Reviewer_PsKo · 2023-11-21

**Novelty:** 5
**Technical Quality:** 5

**Review:**

This paper introduces an anomaly score and simultaneously considers the reconstruction error in the time domain and frequency domain. The authors also use bi-level sliding window techniques to achieve the goal of anomaly detection with fine-resolution among the windows. Overall, this is an interesting paper. However, the paper is very unclear. it is unclear which type of anomalies the proposed method is good at and the experiment seems not to align with the claim in the introduction. The idea of "Time-Frequency Granularity Discrepancy" is not well explained.

**Questions:**

1. The claim seems a little bit contradictory, in Fig. 1 motivation, it seems the authors work on Pattern-Wise anomaly (or discord if considering the definition first introduced in 2005).  But the experiment seems to aim to detect any type of anomalies. And all the anomalies are not well discussed (like why you use a dataset). Given the claim of this paper, I believe universal comparison is insufficient.

2. It seems to introduce a universal anomaly detection approach which claims to be able to detect any type of anomalies. However, it is unclear why t-anomaly + f-anomaly can detect any type of anomalies. I believe authors should analyze how each type of anomalies that could benefit from t-anomaly, f-anomaly, and combined final score in visualization shown in Fig. 6

3. Table 1 information needs more explanation. For example, what is # Point Anomaly Ratio?, # Pattern Anomaly? And their ratio?

4. Most of the example shown in Fig 6 looks like the f-anomaly score plays an important role in anomaly detection (Other than Contextual Point), any example in which t-anomaly score plays an important role in determining the location of the anomaly. Or t-anomaly score is solely used to determine a more concrete location?

5. It would be better to mention the computation cost of the proposed work in anomaly detection.

**Reviewer Confidence:**

3: The reviewer is confident but not certain that the evaluation is correct

**Scope:**

4: The work is relevant to the Web and to the track, and is of broad interest to the community

---

### Official Review · Reviewer_xu3u · 2023-11-21

**Novelty:** 4
**Technical Quality:** 4

**Review:**

This work proposed a unified framework for time series anomaly detection framework called Dual-TF to detect both pattern and point anomalies. The proposed method uses an outer time transformers, and the inner window comparison is achieved by FFT based transformer. Although this paper has a lot of experiments, I have several concerns for this paper.

1. The technique is from two existing framework of attention in time domain and frequency domain, both have been proposed before in other transformer work before. However, simply adding the scores together is not a novel loss function. This is a similar solution just like ensemble, the key discrepancy here is when does pattern based definition useful and when does the point-based definition? The paper did not offer any insight here.
2. The optimal window is determined by dominal frequency and uncertainty, which does not always hold since there can be non-major frequency have anomaly too. For example, in a heartbeat, the meaningful anomaly is usually in subtle changes. The 1/vmajor would simply miss such anomaly.
3. There is a surprising assumption of proof of theorem 3.1 does not make any sense in practice, why the anomaly need to be in a window that is monotonic increasing followed by monotonic decreasing?
4. The efficiency of the method is not mentioned. The two transformers combined would be very expensive to train on large datasets, and could be impossible to deploy in any production situation such as the setting in [3].
5. The literature review is based on very new work in deep learning. However, pattern -based anomaly detection has been done in the last twenty years [1][2], such as context anomaly, collective anomaly, time series discord. There is zero discussion about any of these works.
Although the authors tested one matrix profile, we have no idea on how matrix profile were tested and which parameters it was tested.


[1] Chandola, Varun, Arindam Banerjee, and Vipin Kumar. "Anomaly detection: A survey." ACM computing surveys (CSUR) 41.3 (2009): 1-58.
[2] Keogh, Eamonn, et al. "Finding the most unusual time series subsequence: algorithms and applications." Knowledge and Information Systems 11 (2007): 1-27.
[3] DAMP: accurate time series anomaly detection on trillions of datapoints and ultra-fast arriving data streams. Yue Lu, Renjie Wu, Abdullah Mueen, Maria A. Zuluaga, Eamonn J. Keogh:. Data Min. Knowl. Discov. 37(2): 627-669 (2023)

**Questions:**

1. The technique is from two existing framework of attention in time domain and frequency domain, both have been proposed before in other transformer work before. However, simply adding the scores together is not a novel loss function. This is a similar solution just like the ensemble, the key discrepancy here is simply addressed by a sum, which over-simplifies the problem.
2. The optimal window is determined by dominal frequency and uncertainty, which does not always hold since there can be non-major frequencies have anomalies too. For example, in a heartbeat, the meaningful anomaly is usually in subtle changes. The 1/vmajor would simply miss such an anomaly.
3. There is a surprising assumption of proof of theorem 3.1 does not make any sense in practice, why the anomaly need to be in a window that is monotonic increasing followed by monotonic decreasing?
4. The efficiency of the method is not mentioned. The two transformers combined would be very expensive to train on large datasets, and could be impossible to deploy in any production situation such as the setting in [3].
5. The literature review is based on very new work in deep learning. However, pattern -based anomaly detection has been done in the last twenty years [1][2], such as context anomaly, collective anomaly, time series discord. There is zero discussion about any of these works.
Although the authors tested one matrix profile, we have no idea on how matrix profile were tested and which parameters it was tested.

**Reviewer Confidence:**

3: The reviewer is confident but not certain that the evaluation is correct

**Scope:**

3: The work is somewhat relevant to the Web and to the track, and is of narrow interest to a sub-community

---

### Official Review · Reviewer_ws1N · 2023-11-21

**Novelty:** 3
**Technical Quality:** 5

**Review:**

This paper presents a methodology for anomaly detection in the Time-Series Anomaly Detection task, leveraging both the Time-Domain and Frequency-Domain. It specifically points out the discrepancy in granularity of anomaly scores between the Frequency-Domain and Time-Domain in existing methodologies that use the Frequency-Domain, and proposes a solution to this issue. Overall, the paper strives to validate the proposed method. For example, it explains the importance of utilizing the Frequency-Domain, conducts experiments with appropriate comparison groups (e.g., TFAD, Anomaly Transformer), and demonstrates the effectiveness of the proposed method through an ablation study.

However, the paper could have been improved by more clearly demonstrating the problem caused by the granularity discrepancy and detailing if there are any distinctions beyond the proposed nested sliding window. This would make the Contribution Point of the paper clearer.

**Questions:**

- It seems unclear whether the problem arises from the granularity discrepancy of the Time-Domain and Frequency-Domain scores, or from the coarseness of the Frequency-Domain's Anomaly score.

- The size of the inner-window for the Frequency-Domain is smaller than the size of original window, which seems to limit the context viewed. For example, using the nested-sliding window may be effective for generating fine-grained anomaly scores; however, this approach may limit the consideration of a variety of frequencies. I wonder if there are any considerations for this limitation.

- The approach of creating various window sizes and implementing an anomaly detection method seems not to be substantially different from the proposed approach.

**Reviewer Confidence:**

4: The reviewer is certain that the evaluation is correct and very familiar with the relevant literature

**Scope:**

4: The work is relevant to the Web and to the track, and is of broad interest to the community

---

### Official Review · Reviewer_gMPs · 2023-11-26

**Novelty:** 6
**Technical Quality:** 6

**Review:**

This paper proposes a new approach for time-series anomaly detection using a nested time window to cover the time and frequency domains and overcome the time-frequency granularity discrepancy in anomaly detection.
The authors address a principal research problem in the time-series anomaly detection area that is highly relevant to The Web conference community.
The paper explains well why this research problem is an important problem for many Web applications in the introduction section.
The related work sections describe well the state-of-the-art work regarding time series anomaly detection and frequency domain analysis for time series.

The proposed approach is very well described and formulated. However, some of the formulation is not easy to follow as it uses its unique formulation. I suggest adding some text to describe some essential steps in more detail.
Evaluation is done on multiple datasets and provides detailed comparisons with the related stat-of-the-art approaches for TSAD.

**Questions:**

How important is the size of the inner window in comparison to the outer window in Dual-TF in terms of the percentage of the outer window? Are there any of the evaluation results that experiment with the window sizes? For example, a graph that shows the F1 score by changing the outer and inner window size?

**Ethics Review Description:**

Public datasets are used for evaluation

**Reviewer Confidence:**

3: The reviewer is confident but not certain that the evaluation is correct

**Scope:**

3: The work is somewhat relevant to the Web and to the track, and is of narrow interest to a sub-community

---

### Official Review · Reviewer_NKfj · 2023-11-29

**Novelty:** 4
**Technical Quality:** 5

**Review:**

The paper starts by introducing the time-frequency granularity discrepancy for time-series anomaly detection. A windowing mechanism is employed to resolve this discrepancy and generate anomaly scores from both domains. The end-to-end framework consists of two Transformer architectures. Experimental results on several datasets demonstrate the improved performance from the proposed solution.

Strengths:

- Nicely motivated discrepancy for time-series anomaly detection
- Well-written paper and good coverage of related work
- Experimental results support the overall claims of the paper over multiple datasets

Weaknesses:

- It's not clear why such a complex solution is necessary. We have methods working on time domain. We have methods working in the frequency domain. Then, the anomaly scores (all in the time domain) can be averaged for example, to detect anomalies. A simple baseline like that is necessary to understand (and ensure) that this is indeed a complex problem and such solutions are ineffective. To state that differently, it's not clear if the improvement is coming due to technical contribution or because more "data angles/representations" are used (i.e., you have an ensembling solution but you do not compare against other ensembling solutions, a simple averaging of the score can reveal the strength of multiple methods).

- The paper cites a recent benchmark, TSB-UAD [6], but does not compare against all datasets (2000 time series) and all baselines. Hence, it's not clear if the proposed solution really advances the state of the art in the area.

- The parameter settings are not clear. In some methods you mention that parameters were chosen from a set of values (how?) in other you say you use the default parameters in these methods (is that fair?). In unclear if all these methods were compared fairly, under the same settings or just results were taken "as is" from previous papers.

- The results are presented in a strange way. First the F1 measure is presented, then Range-AUC and VUS separately but only for 2 methods. Then ablation study happens with F1 again. Why don't you present all measures in all tables to easy identification of patterns

- The paper seems to be focusing mainly on DNN solutions. Data mining methodologies are only evaluated in the appendix, which is not ideal. Several methods are missing, some examples (and there are much more in the literature).

"Series2graph: Graph-based subsequence anomaly detection for time series." arXiv preprint arXiv:2207.12208 (2022).
"SAND: streaming subsequence anomaly detection." Proceedings of the VLDB Endowment 14.10 (2021): 1717-1729.

**Questions:**

Relevance to WWW is not strong. An alternative venue might be more appropriate for this line of work (nothing in the paper has to do with Web data).

Please respond to the above comments.

**Reviewer Confidence:**

4: The reviewer is certain that the evaluation is correct and very familiar with the relevant literature

**Scope:**

1: The work is irrelevant to the Web

---

### Decision · Program_Chairs · 2024-01-22

**Decision:**

Accept

**Comment:**

This is the meta-review by the SPC responsible for your paper, and takes into account the opinions expressed by the referees, the subsequent decision thread, and my own opinions about your work.

 - This paper presents a framework called Dual-TF for anomaly detection in the time-series anomaly detection task. A sliding window mechanism is introduced to resolve the discrepancy in the time and frequency domains and generate anomaly scores.

 - The reviewers expressed some concerns about the motivation, efficiency, unclear presentation, etc., while the authors responded to these concerns with detailed explanations and more supported results, which improved the soundness and comprehension of the paper, and committed to resolving in the final version.